# Phosphatase specificity influences phosphorylation timing of CDK substrates during the cell cycle

Theresa U. Zeisner [1,4] ✉, Tania Auchynnikava[1,2], Emma L. Roberts [1] & Paul Nurse [1,3]

Cell cycle events are ordered by cyclin-dependent kinases (CDKs), which phosphorylate hundreds of substrates. Multiple phosphatases oppose these CDK substrates, yet their collective role in regulating phosphorylation timing in vivo remains unclear. Here, we show that four phosphatases (PP2A-B55, PP2A-B56, CDC14, and PP1) each target distinct subsets of CDK substrate sites in vivo in fission yeast, influencing when phosphorylation occurs during G2 and mitosis. On average, sites dephosphorylated by CDC14 and PP2A-B56 are phosphorylated earlier during G2, followed by sites dephosphorylated by PP1 and PP2A-B55. This suggests that these phosphatases set different phosphorylation thresholds at the G2/M transition. Consistent with this, depleting PP2A-B55 or CDC14 accelerates mitotic onset, likely by advancing phosphorylation of their respective CDK substrates, suggesting these phosphorylation thresholds are important for regulating mitotic onset. Our findings establish in vivo phosphatase substrate specificity as a key factor regulating the timing of CDK substrate phosphorylation throughout the cell cycle.

The events of the cell cycle are driven by the reversible phosphorylation of hundreds of CDK substrates. These phosphorylations must occur in the correct order to ensure proper cell cycle progression and viable daughter cells[1]. A core mechanism to achieve this order is the in vivo sensitivity of CDK substrate sites to increasing CDK activity[2,3]. CDK activity is low during G1 and S-phase, resulting in the phosphorylation of S-phase but not mitotic substrates[2]. Dephosphorylation of CDK-Tyr15 at the G2/M transition results in a switch-like increase in CDK activity, leading to the phosphorylation of CDK substrates at mitosis[4,5]. However, the efficiency with which CDK substrates are phosphorylated by CDK in vitro does not correlate with the timing of their phosphorylation during the cell cycle[6]. This suggests that, in vivo, additional factors, such as opposing phosphatase activity, also impact the timing of CDK substrate phosphorylation. Here, we investigate CDK-opposing phosphatases to determine whether they influence the timing of CDK substrate phosphorylation. Multiple phosphatases have been reported to target CDK substrates[7,8], but for the majority of these phosphosites, it is not known which phosphatase targets them, and a systematic understanding of how these phosphatases together shape the phosphorylation timing of CDK substrates during interphase is currently lacking.

Genetic screens, biochemical and cell-based assays have shown that the CDK-opposing phosphatases, PP2A-B55, PP2A-B56, PP1 and CDC14, are important cell cycle regulators, affecting processes such as chromosome biorientation, the spindle assembly checkpoint, and mitotic entry in a range of organisms[7,8]. Recent biochemical work has revealed that specific substrate targeting mechanisms are employed by these phosphatases, including preferences for the phosphorylated residue and the surrounding amino acids as well as substrate recruitment via binding interactions[9–12]. These results suggest that many phosphosites are dephosphorylated by a single phosphatase, but most of these studies investigated individual phosphatases in isolation. In

[1]Cell Cycle Laboratory, The Francis Crick Institute, London, UK. [2]Proteomics Platform, The Francis Crick Institute, London, UK. [3]Laboratory of Yeast Genetics and Cell Biology, Rockefeller University, New York, NY, USA. [4]Present address: Research Institute of Molecular Pathology (IMP), Vienna BioCenter (VBC), 1030 Vienna, Austria. ✉e-mail: theresa.zeisner@imp.ac.at

contrast, in vivo work from budding yeast investigating PP2A-B55, PP2A-B56 and CDC14 suggests that at mitotic exit CDK substrates are jointly targeted by these three phosphatases[13]. Thus, it remains unclear whether specific CDK substrate sites are targeted by individual CDK-opposing phosphatases or jointly by multiple phosphatases in vivo.

A further unanswered question is what effect phosphatases may have on the timing of CDK substrate phosphorylation during the cell cycle. So far, CDK-opposing phosphatases have been studied in-depth for their crucial roles during mitosis and at mitotic exit through dephosphorylating CDK substrates[13–17]. However, these phosphatases are not just active at mitotic exit and can continuously remove phosphate groups from CDK substrate sites during the rest of the cell cycle. The overall amount of phosphorylated substrate (net phosphorylation) is determined by the catalytic activities of both the kinase and the opposing phosphatase. A rapid phosphatase-dependent turnover of a CDK substrate site will set a higher threshold that CDK activity must reach to bring about net phosphorylation of such a site, thus delaying net phosphorylation until later during the cell cycle when CDK activity is higher[18]. Consistent with this, inhibition of PP2A-B55 in budding yeast advances the net phosphorylation of threonine-containing sites and the CDK substrate Ndd1[19]. However, in budding yeast, it has also been concluded that the impact of phosphatases does not predict the timing of CDK substrate phosphorylation[20]. Thus, how different CDK-opposing phosphatases affect the timing of substrate phosphorylation during interphase also remains contentious.

Given these unanswered questions, we have identified in vivo substrate sites of four CDK-opposing phosphatases: PP2A-B55, PP2A-B56, PP1 and CDC14. We show that these four phosphatases mostly target distinct subgroups of CDK substrate sites, which are, on average, net phosphorylated at different times during the cell cycle. This suggests that there is a division of labour between the phosphatases,

which impacts the timing of CDK substrate phosphorylation during G2, onto the onset of mitosis.

## Results

### Identification of CDK-dependent phosphatase substrates in vivo

We identified which phosphosites on CDK substrates (hereafter referred to as CDK substrate sites) are targeted by PP2A-B55, PP2A-B56, PP1, and CDC14 in vivo using phosphoproteomics by comparing the dephosphorylation kinetics of CDK substrate sites in the presence and absence of each phosphatase. Fission yeast has a single CDK encoded by the *cdc2* gene. To identify CDK substrate sites, we synchronised cells in G2 with an ATP analogue-sensitive CDK allele (*cdc2-as*) and released them into mitosis, at which point CDK substrate sites are highly phosphorylated[21,22]. The dephosphorylation of phosphosites upon CDK inhibition with the ATP analogue 1-NmPP1 was monitored using phosphoproteomics and confirmed by western blotting using a phospho-TPxK antibody (Fig. 1a and Supplementary Fig. 1a–c). We classified phosphosites as CDK substrate sites if they were dephosphorylated rapidly upon CDK inhibition by fitting an exponential decay curve (Methods, Supplementary Fig. 1d–g). The identified CDK substrate sites were enriched for the full CDK consensus motif (S/TPxK/R) and for gene ontology terms associated with the cell cycle[23] (Supplementary Fig. 1h, i). Consistent with previous data, CDK substrate sites had a median half-life of <2 mins, whilst protein levels remained stable (Supplementary Fig. 1j, k)[2]. Around 15% of CDK substrate sites had a half-life of less than 30 s, emphasising the rapid turnover of these phosphosites (Supplementary Data 1). The dephosphorylation rates of CDK substrate sites were similar between the four phosphoproteomic experiments, indicating reproducibility between biological repeats (Supplementary Fig. 1l).

The rapid dephosphorylation of CDK substrate sites indicates that phosphatases are active. We, therefore, expected that in the absence of

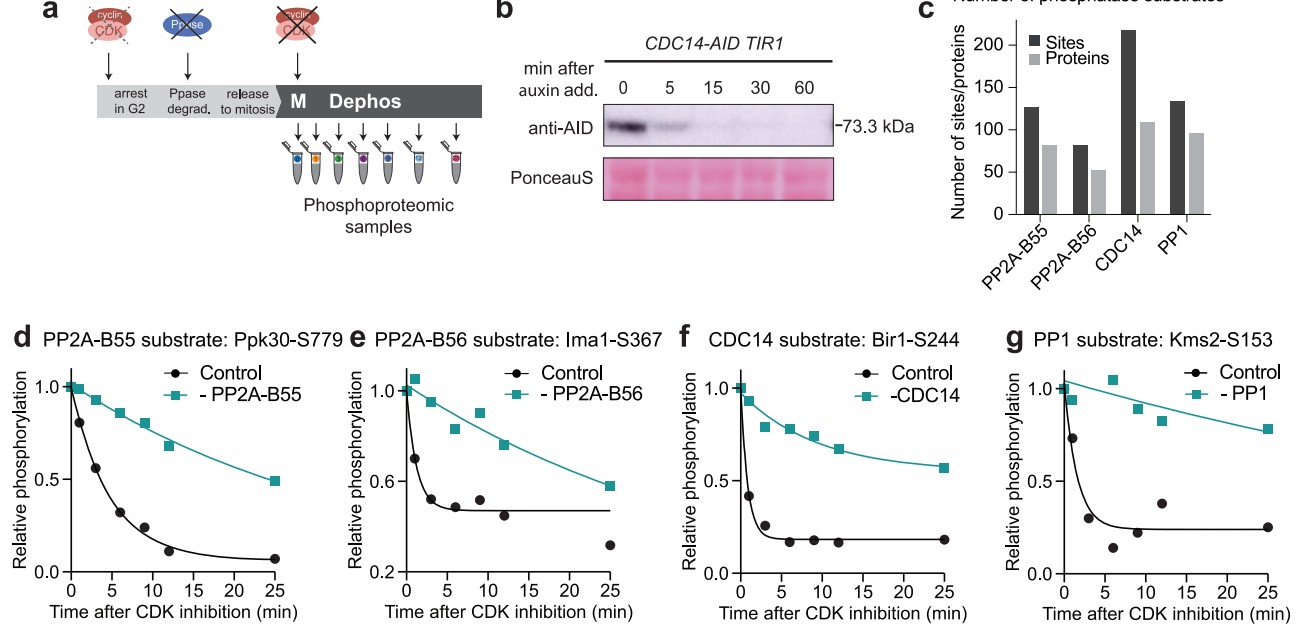

**Fig. 1 | Identification of in vivo CDK-dependent phosphatase substrates.**
**a** Schematic of the experimental set-up to identify phosphatase substrates in a synchronised culture upon CDK inhibition. Initially, 1 μM 1-NmPP1 was used to arrest cells in G2. Auxin was added to G2-arrested cells to degrade the phosphatases of interest. Cells were released into mitosis by washing with 1-NmPP1 free media 10 mins after auxin addition. At peak mitosis, 8 mins after release, CDK activity was inhibited fully by adding 10 μM 1-NmPP1. **b** Western blot against the AID-tag of CDC14 (73.3 kDa), showing degradation upon auxin addition. Ponceau-S stain was used for total protein normalisation. The uncropped blot is provided in the Source

data, *n* = 1. Rapid degradation of phosphatases upon auxin addition was further confirmed by Western blots shown in Supplementary Fig. 1c. **c** Grouped bar graphs showing the number of CDK-dependent phosphatase substrate sites (dark grey) and proteins (light grey). **d–g** Examples of identified phosphatase substrates. Relative phosphorylation level (phosphorylation normalised to the respective timepoint 0) in the presence (black) and absence (turquoise) of phosphatase of interest upon CDK inhibition. Curves are a one-phase exponential decay fitted to the relative phosphorylation level determined by phosphoproteomics.

the opposing phosphatase, the dephosphorylation of a site to be slower. Thus, to identify which phosphatases dephosphorylate which specific CDK substrate sites, we quantified the dephosphorylation of CDK substrate sites in the absence of each of the four phosphatases. Using an auxin-inducible degron (AID) system, we individually depleted the activity of the four CDK-opposing phosphatases (Fig. 1b, Supplementary Fig. 1a, b)[24,25]. PP2A and PP1 are both holoenzymes consisting of different subunits (Supplementary Fig. 2a). For PP2A, we individually degraded the regulatory subunits B55 (encoded by *pab1*) and B56 (encoded by *par1* and *par2*), thus abolishing the activity of two specific PP2A holoenzymes separately. PP1 associates with multiple regulatory subunits; for example, there are >200 regulatory PP1 subunits in human cells[26]. We, therefore, degraded both catalytic subunits of PP1 together (encoded by *dis2* and *sds21*), thereby inhibiting the activity of all possible PP1 holoenzymes. We also degraded CDC14 (encoded by *clp1*), which does not form a holoenzyme. Degradation of the phosphatases was rapid, occurring within 15 min and led to phenotypes similar to previously reported deletion phenotypes of the corresponding genes (Fig. 1b and Supplementary Fig. 2b, Supplementary Fig. 3 a, b). Deletion of both catalytic subunits of PP1 is lethal. Degrading both subunits stopped length extension, yet cells still progressed through the cell cycle until metaphase, where they arrested (Supplementary Movies 1&2, Supplementary Fig. 2c).

We classified CDK substrate sites as phosphatase targets if their dephosphorylation was substantially slower in the absence of one of the phosphatases by comparing the dephosphorylation in the control and phosphatase-degraded conditions (Methods and Supplementary Fig. 1d–e). We identified between 80-220 CDK-dependent phosphatase substrate sites for each of the four investigated phosphatases, together totalling 542 sites on 262 proteins (Fig. 1c–g and Supplementary Fig. 4a, b and Supplementary Data 1). For each of the investigated phosphatases, our dataset included phosphatase substrates that have been previously identified, such as the PP2A-B55 substrate PRC1 and the CDC14 substrate MSH6[10,27] (Supplementary Data 1). We also verified our phosphoproteomic data for two example substrate sites with western blotting, with phosphospecific antibodies against Spo15-pS121 and Cut12-pT75 (Supplementary Fig. 4c–h). Overall, this approach provides a comprehensive dataset of CDK-dependent and, thus, cell cycle-relevant phosphatase substrates.

## Phosphatases target distinct subgroups of CDK substrate sites

Our dataset enabled us to determine if one or multiple phosphatases dephosphorylated a specific CDK substrate site (Fig. 2a and Supplementary Fig. 5a–c). To allow comparisons across all datasets, only sites detected in all four experiments were used for this analysis (274 sites) (Supplementary Fig. 5d, e). We identified the opposing phosphatase for 59% of these common CDK substrate sites (Fig. 2b). There was very little overlap between the groups of phosphatase substrates, with less than 5% of CDK substrate sites being dephosphorylated by multiple phosphatases (Fig. 2b and Supplementary Fig. 5f–i). This indicates that these phosphatases target distinct substrate pools in vivo, in contrast to data from budding yeast suggesting that PP2A-B55, PP2A-B56 and CDC14 jointly target their substrates[13]. This division of labour between phosphatases in opposing CDK substrate sites raises two questions: what is the molecular basis for this apparent substrate specificity, and what regulatory consequences does this division of labour have for the net phosphorylation timing of CDK substrate sites.

We first assessed whether phosphatase specificity arises at the protein or phosphosite level, by examining proteins with multiple CDK substrate sites. For 76% of multiple-phosphorylated CDK substrate proteins, different phosphosites on the same protein were targeted by different phosphatases (Fig. 2c). This indicates that phosphatase substrate specificity operates at the level of individual phosphosites rather than the level of the entire protein. To interrogate the molecular basis of this, we investigated whether sites targeted by a given phosphatase

shared common features. Comparing the sequences of the phosphatase substrate sites with that of all CDK substrate sites revealed differences in the amino acids surrounding phosphorylation sites (Fig. 2d–g, Supplementary Fig. 6a, b). PP2A-B56 substrates were enriched for basic amino acids surrounding the phosphorylation site (Fig. 2g), and CDC14 substrates were enriched for phosphoSerine (pSerines) and lysine at the +3 position, which corresponds to the full CDK consensus motif[23] (Fig. 2f). PP1 substrates were strongly enriched for phosphoThreonine (pThreonines) at the phosphorylation site (Fig. 2h), even though around half of CDK-dependent PP1 substrates are pSerines. This can be explained by the higher abundance of serines than threonines in the phosphoproteome[28] (Supplementary Fig. 6c). This result emphasises the need to identify individual substrates of phosphatases experimentally under in vivo conditions, rather than relying solely on consensus motifs for their identification. Consistent with previous work, PP2A-B55 and PP1 dephosphorylated pThreonine CDK substrate sites significantly faster than pSerine sites[11,12,29] (Supplementary Fig. 6d–g).

The identified phosphatase substrates highlight the regulatory interplay between CDK and phosphatases, as well as between the phosphatases themselves. We measured CDK-Tyr15 phosphorylation level as a proxy for CDK activity, showing that CDK activity was higher in the absence of PP2A-B55, PP2A-B56, CDC14, and to a lesser extent in the absence of PP1 (Supplementary Fig. 7a, b). This negative regulatory effect is likely at least partly due to these phosphatases targeting different sites on the CDK-activating phosphatase Cdc25. We further identified a known CDK-substrate site on PP1, which negatively regulates PP1 activity (Dis2-T316), and identified a PP1 site on Polo-kinase. We also showed that PP2A-B56 dephosphorylates a CDK substrate site on CDC14 (Supplementary Fig. 7c, d, Supplementary Data 1).

We conclude that CDK-dependent phosphatase substrates are site-specific rather than protein-specific, and site-specificity is at least partly determined by the different amino acid preferences of each phosphatase.

## Identification of CDK-independent phosphatase substrates reveals conservation of phosphatase substrate motifs

The investigated phosphatases are also known to dephosphorylate sites, not phosphorylated by CDK, and we therefore extended our analysis to identify which non-CDK substrates are targeted by the four phosphatases[8]. We reasoned that the phosphorylation levels of sites phosphorylated by kinases other than CDK would remain unchanged immediately after inhibiting CDK activity in mitosis. We used the same phosphoproteomic dataset to identify sites where the phosphorylation level remained unchanged upon CDK inhibition, classifying them as CDK-independent sites (Fig. 3a, b and Supplementary Fig. 8a, b). CDK-independent sites for which the phosphorylation level was significantly higher in the absence of the phosphatase were classified as CDK-independent phosphatase substrates (Fig. 3a, b and Supplementary Fig. 8a). This analysis identified between 600-1300 phosphatase substrates for each of the four investigated phosphatases, totalling 3400 sites (Supplementary Fig. 8b–d, Supplementary Data 2). Combining the CDK-dependent and CDK-independent phosphatase substrates together gave an estimate of the total substrate sites targeted by each phosphatase (Fig. 3c). As before, to compare between datasets, we only included sites detected in all four experimental sets. This showed, that similar to the CDK-dependent phosphatase substrates, there was limited overlap between the different groups of total phosphatase substrates, with less than 3% of sites targeted by multiple phosphatases (Fig. 3d). We estimated that PP1 and PP2A-B55 each targeted approximately 6-9% of all phosphosites in the phosphoproteome, while PP2A-B56 and CDC14 each targeted an additional 3-4% of all phosphosites, bringing the total to around 25% of the entire phosphoproteome (Fig. 3d).

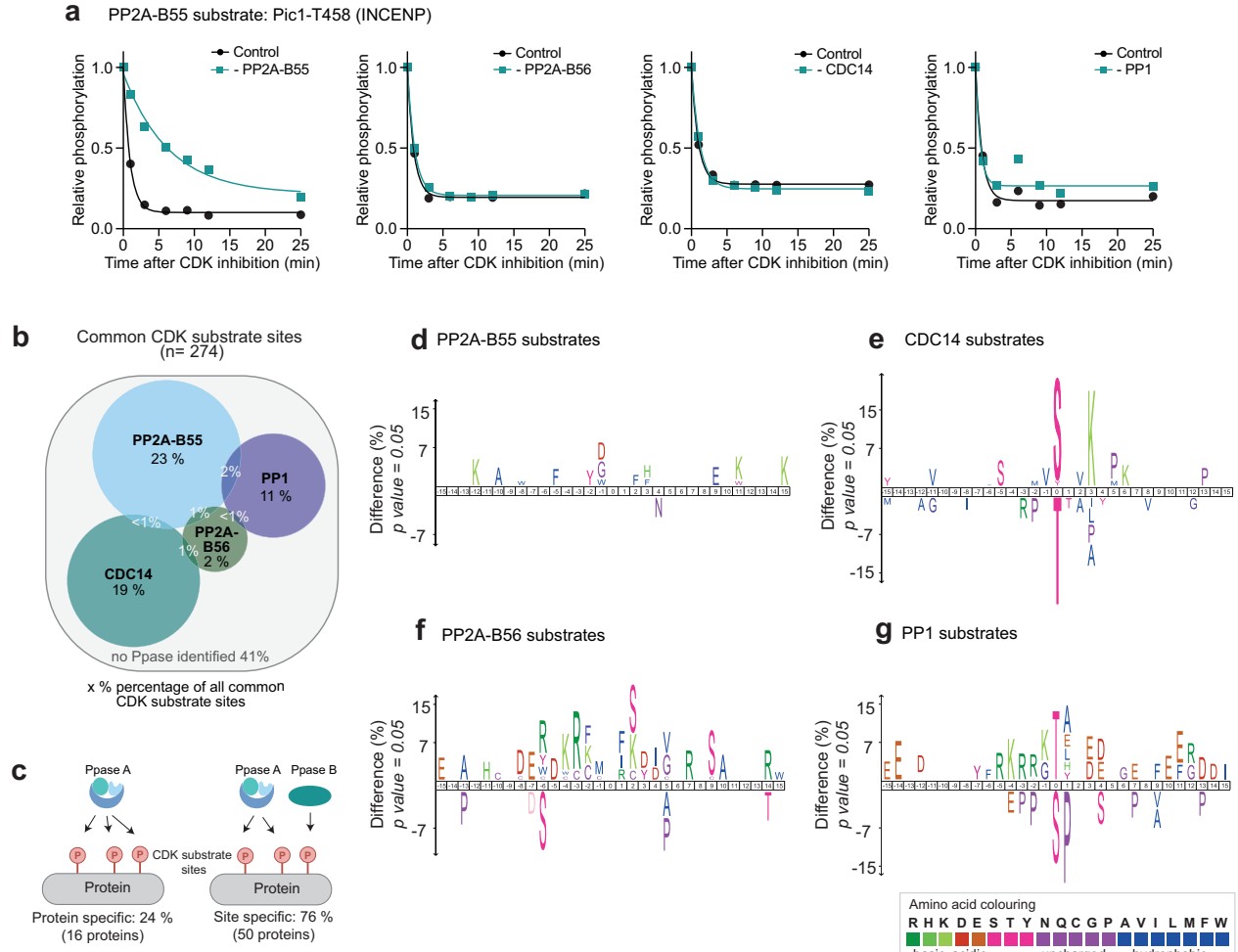

**Fig. 2 | Phosphatases target distinct subgroups of CDK substrate sites.**
**a** Example of an identified PP2A-B55 phosphatase substrate. Relative phosphorylation level in the presence (black) and absence (turquoise) of phosphatase of interest upon CDK inhibition. Curves are a one-phase exponential decay fitted to the relative phosphorylation level determined by phosphoproteomics. **b** Venn diagram depicting the overlap of CDK-dependent phosphatase substrates of PP2A-B55, PP2A-B56, PP1 and CDC14 (using only CDK-dependent sites common to all four experimental sets as background) **c** Schematic of protein-specific or site-specific

phosphatase substrates on proteins with multiple CDK-dependent sites. **d**–**g** Amino acid motifs of CDK-dependent phosphatase substrates of the four different phosphatases. A 31 amino acid sequence centred on the phosphorylated residue of the identified phosphatase substrates was compared against all identified CDK substrate sites using iceLogo (all changes shown are significant with $p$-value < 0.05, two-tailed Student's t-test)[69]. Amino acids above the line are enriched, while those below the line are depleted in the phosphatase substrates.

Taking the CDK-dependent and CDK-independent phosphatase substrates together revealed distinct motifs of amino acids surrounding the phosphorylated residue for PP2A-B55, PP2A-B56, PP1, and CDC14 substrates (Fig. 3e–h, Supplementary Fig. 8f, g). PP2A-B55 substrates were enriched for pThreonine followed by a Proline, while these residues were depleted in PP2A-B56 substrates (Fig. 3e, g). PP2A-B56 substrates were strongly enriched in basic amino acids at the −2 and −3 position, which resembles the Aurora B kinase motif[30]. CDC14 substrates were enriched for PxK (+1 to +3 position), consistent with the full CDK consensus motif[23]. PP1 substrates were enriched for pThreonines, basic amino acids upstream of phosphorylation sites, and depleted for prolines at the +1 position. CDK also phosphorylates non-proline sites (non-canonical CDK substrate sites)[31] and consistent with this, approximately 40% of CDK-dependent PP1 substrates were non-canonical CDK substrate sites (Supplementary Data 1). Overall, the phosphatase motifs showed a high similarity with motifs of metazoan phosphatases[9–12], indicating conservation of substrate specificity across eukaryotes.

To investigate which kinases these phosphatases oppose in addition to CDK, we conducted an enrichment analysis based on the

kinase substrate motifs from the human kinome atlas[32]. This revealed that PP2A-B55 substrates were enriched for MAPK motifs and, as expected, CDK kinase motifs, while PP2A-B56 substrates were strongly enriched for Aurora kinase motifs[30] (Supplementary Fig. 9a). Additionally, we also searched the groups of total phosphatase substrates for consensus motifs of cell cycle-relevant kinases, including CDK, Aurora, Polo, NIMA, SID2 and DDK[23,30,33–37]. This confirmed the preference of PP2A-B56 for Aurora substrates and of PP2A-B55 and CDC14 for CDK substrates. It further emphasised that each of the phosphatases targets a range of cell-cycle kinase substrates (Supplementary Fig. 9b–c). In addition to preferences for amino acids surrounding the phosphorylated residue, phosphatases can further enhance substrate specificity by binding to short linear motifs (SLiMs)[38]. PP2A-B56 substrates were enriched for LxxIxE motifs known to interact with a conserved region on B56, while CDC14 substrates were enriched for PxL motifs (Supplementary Fig. 9d)[39,40]. We also conducted a motif enrichment analysis using the MEME tool suite SEA (simple enrichment analysis)[41]. This identified known motifs that are relatively enriched in the total phosphatase substrate sequences and yielded between 7 and 23 motifs (5-15 amino acids long) for each phosphatase substrate

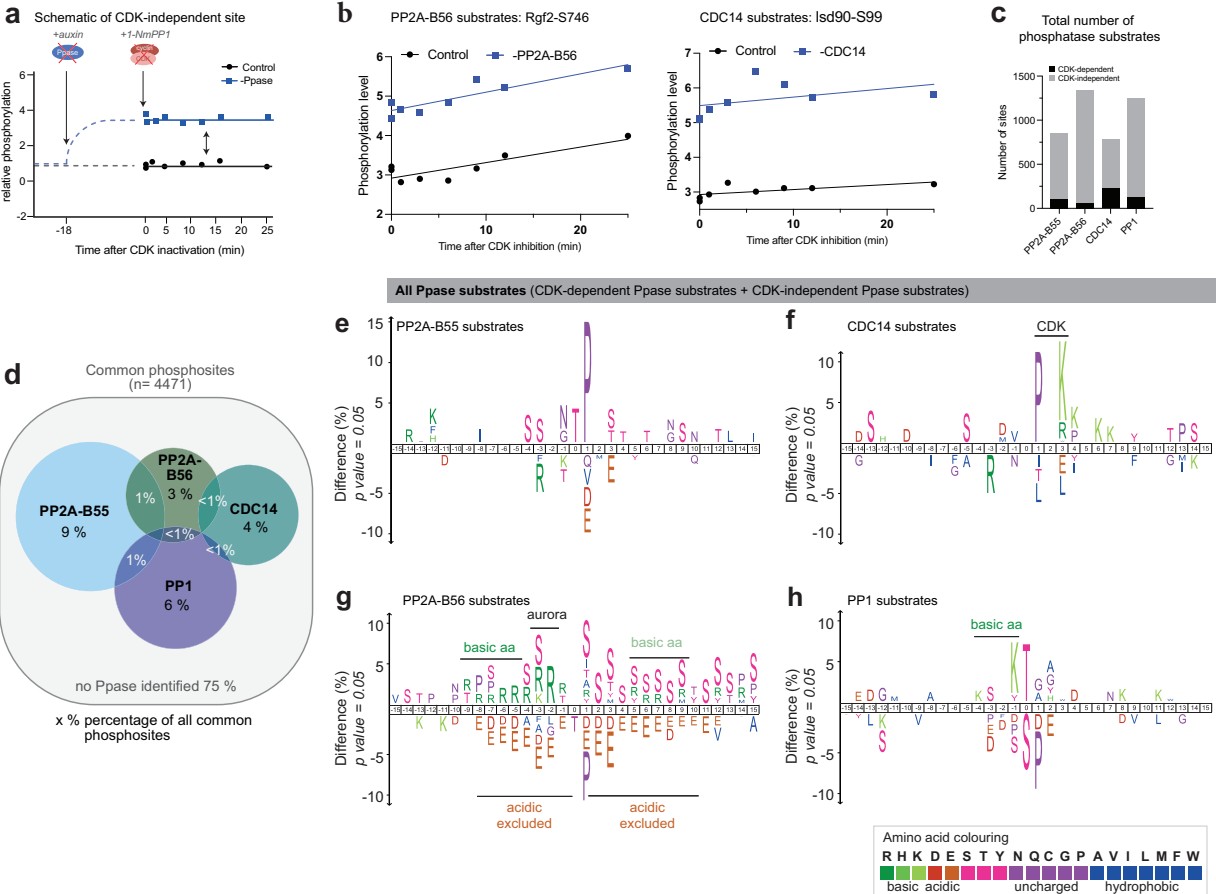

**Fig. 3 | Phosphatases target distinct subgroups of CDK-independent sites.**
**a** Schematic of the relative phosphorylation level of a CDK-independent phosphatase substrate in the presence (black) and absence (blue) of the phosphatase. **b** Examples of CDK-independent substrates of PP2A-B56 and CDC14. Unlogged phosphorylation level in the presence (black) and absence (blue) of the indicated phosphatase of interest upon CDK inhibition is shown. Lines are linear regression fitted to the unlogged phosphorylation level determined by phosphoproteomics. **c** Stacked bar graph showing the total number of all CDK-dependent and independent phosphatase substrates (not normalised to the total number of detected sites per experiment). **d** Venn diagram depicting the overlap of total phosphatase substrates, using only phosphosites common to all four experimental sets. **e–h** Amino acid motifs of total phosphatase substrates (CDK-dependent + CDK-independent phosphatase substrates) of the four investigated phosphatases. A 31 amino acid sequence centred on the phosphorylated residue of the identified phosphatase substrates was compared against all identified phosphosites sites using iceLogo (all changes shown are significant with p-value < 0.05, two-tailed Student's t-test)[69]. Amino acids above the line are enriched, while those below the line are depleted in the phosphatase substrates.

group (Supplementary Data 3). Interestingly, PP2A-B56 substrate sites were enriched for SILK motifs, a known PP1 SLiM, suggesting that PP2A-B56 substrate sites may be in close proximity to PP1 binding sites.

This analysis revealed that the phosphatases target substrates of multiple cell cycle protein kinases, including Polo and Aurora, and emphasises that kinases and phosphatases generally do not act in pairs; rather, a single phosphatase targets substrates from several different kinases. Together, these data further confirm that the investigated phosphatases are specific enzymes which target distinct groups of substrates in vivo.

## CDK substrates opposed by different phosphatases are net phosphorylated at different cell cycle times

We next investigated whether the division of labour between phosphatases in opposing distinct subgroups of CDK substrate sites had a functional impact on the timing of CDK substrate phosphorylation. In particular, we analysed a phosphoproteomic timecourse dataset to determine whether the net phosphorylation timing of CDK substrates differed between the groups of phosphatase substrates. This dataset quantified relative CDK substrate phosphorylations during a synchronised cell cycle following release from a G2 arrest in the presence

of all phosphatases (Supplementary Fig. 10a)[2]. Comparing the median relative phosphorylation profiles of CDK substrate sites dephosphorylated by different phosphatases revealed a temporal order during the cell cycle (Fig. 4a, Supplementary Fig. 10b). CDK substrate sites targeted by PP2A-B56 and CDC14 were, on average, net phosphorylated earlier during the cell cycle, followed by CDK substrate sites targeted by PP1 and then PP2A-B55. The median relative phosphorylation of CDK/CDC14 sites started to increase around 20 min earlier than CDK/PP2A-B55 sites (Fig. 4a).

To quantify the phosphorylation timing of individual CDK substrate sites, we calculated the area under the curve (AUC) of the relative phosphorylation level for each CDK substrate from G2 phase to mitosis (50-100 mins after release from a G2 arrest). The later a substrate is phosphorylated, the smaller is its AUC (Supplementary Fig. 10c). This analysis revealed a spread of phosphorylation timings within each group of phosphatase substrates (Fig. 4b). The median AUC values confirm the earlier finding that, on average, CDK substrate sites targeted by PP2A-B56 and CDC14 were net phosphorylated earlier (higher median AUC), while sites targeted by PP2A-B55 and PP1 were net phosphorylated significantly later (lower median AUC). It also demonstrated that S-phase substrates as a whole were targeted by

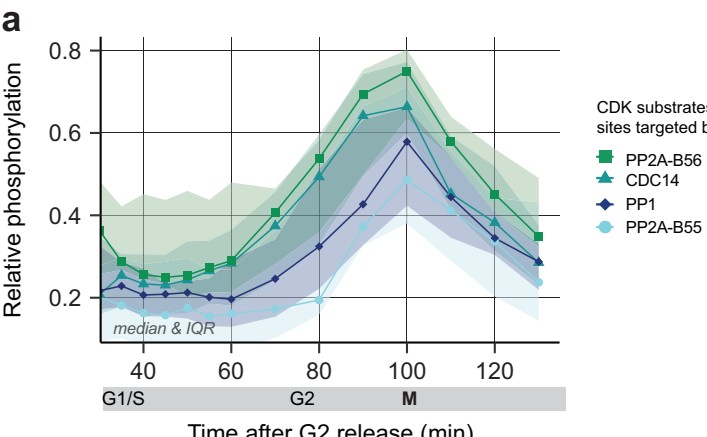

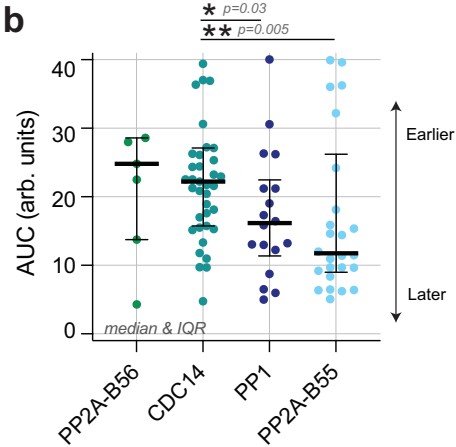

**Fig. 4 | Net phosphorylation timing of CDK substrate sites differs between phosphatase substrate groups. a** Median of relative phosphorylation of CDK-dependent phosphatase substrates during a synchronised cell cycle, identified by mass spectrometry. Relative phosphorylation refers to the phosphorylation level relative to the phosphorylation level at mitosis. Lines denote medians and ribbons inter-quartile range (IQR). **b** Beeswarm plot of AUC values of CDK-dependent phosphatase substrate sites, split according to which phosphatase dephosphorylates the CDK substrate site (PP2A-B56, $n = 7$; CDC14, $n = 39$; PP1, $n = 18$; PP2A-B55, $n = 26$). AUC values are the integral of the phosphorylation level (50-100 mins after release from G2) and act as a proxy for phosphorylation timing and pattern during the cell cycle. Error bars correspond to median and IQR. Significant differences were determined using the Mann-Whitney test (two-tailed).

multiple, rather than just one, phosphatase (Supplementary Data 1). We conclude that the identity of the phosphatase that dephosphorylates a CDK substrate appears to impact the net phosphorylation timing of CDK substrates during G2 and mitosis but does not distinguish between S-phase and mitotic CDK substrates.

There is a considerable spread in the timing of CDK substrate phosphorylations within each group of phosphatase substrates (Fig. 4b). This indicates that the exact net phosphorylation timing of each CDK substrate site can be expected to be the consequence of several factors, including how well a site is phosphorylated by CDK, the identity of the opposing phosphatase, the presence of SLiMs, and the sub-cellular localisation of the substrate[18,42,43]. To stratify the contributions of some of these factors, we grouped CDK substrates according to these criteria (Supplementary Fig. 10d). By comparing the difference of median AUCs between sub-categories of each of these factors, we can estimate the size of their effects. This showed that the identity of the phosphatase and substrate localisation had the greatest difference in AUC values between groups, being 3.5 and 3 times larger compared to AUC differences seen when CDK substrates are grouped according to the phosphorylated residue (Supplementary Fig. 10d). This indicates that both the identity of the phosphatase and sub-cellular localisation of the CDK substrates have strong effects on the cell cycle timing of net CDK substrate phosphorylation. The observed differences in the net phosphorylation timing of CDK substrates, categorised by phosphatases or subcellular location, are independent. If these two categories were connected, CDC14 and PP2A-B56 substrates, which are net phosphorylated first during the cell cycle, should be predominantly located in the nucleus, as that is the compartment in which CDK activity rises first[44]. However, none of the phosphatase substrate groups were significantly enriched in any specific subcellular compartment (Supplementary Fig. 10e). Furthermore, if we restricted sites to a specific sub-cellular compartment, such as the nucleus or cytoplasm, CDC14 substrate sites were still net phosphorylated before PP2A-B55 sites, regardless of the cellular compartment (Supplementary Fig. 10f).

We conclude that the division of labour between the phosphatases that dephosphorylate distinct subgroups of CDK substrate sites plays a role in ordering the net phosphorylation of CDK substrate sites during the transition from G2 into mitosis. Our data showed that CDK substrate sites opposed by CDC14 were, on average, phosphorylated earlier than those opposed by PP1 and PP2A-B55.

## PP2A-B55 impacts the phosphorylation timing of a CDK/PP2A-B55 substrate in vivo

How efficiently a CDK substrate is dephosphorylated by a given phosphatase may affect net phosphorylation timing within the cell cycle[18]. Since CDK substrate sites targeted by PP2A-B55 were, on average, net phosphorylated significantly later than the average CDK substrate site during G2, we interrogated the possibility that PP2A-B55 may delay net phosphorylation of its CDK substrate sites. To do so, we used an in vivo phosphorylation-sensor to determine whether a PP2A-B55 site is net phosphorylated earlier during the cell cycle in the absence of the phosphatase. We identified Cut3-T19, which is part of the in vivo phospho-sensor SynCut3-mCherry, as a PP2A-B55 substrate site in our phosphoproteomic dataset and confirmed a direct interaction between Cut3 and PP2A by co-immunoprecipitation (Fig. 5a, b, Supplementary Fig. 11a). Upon net phosphorylation of T19 by CDK, Cut3 translocates from the cytoplasm into the nucleus at the onset of mitosis[45], so the nuclear-to-cytoplasmic ratio (N/C) of exogenously expressed SynCut3-mCherry can be used to monitor its phosphorylation status (Fig. 5c). In the presence of PP2A-B55 the net phosphorylation of T19 is low during interphase and abruptly increases around 12 min before peak mitosis. In the absence of PP2A-B55, the net phosphorylation of T19 begins earlier in G2, on average 24 min before the mitotic peak, and its initial net phosphorylation increases less rapidly (Fig. 5d, e and Supplementary Fig. 11b). Net phosphorylation timing is also advanced when B55 is deleted rather than degraded (Supplementary Fig. 11b-d). This change in SynCut3-T19 phosphorylation pattern is not observed in the absence of other cell cycle phosphatases (CDC14, Pyp1, PP1 catalytic subunit), indicating that this is a phosphatase-specific effect (Supplementary Fig. 11e-g).

To confirm this phosphatase-specific effect further, we deleted the major catalytic subunit of the PP2A holoenzyme (*ppa2*), which partially reduces PP2A-B55 activity. As expected, SynCut3-T19 is net phosphorylated marginally earlier in the absence of the major PP2A catalytic subunit (*ppa2*) (Supplementary Fig. 11e-g). Mutating the phosphosite of the sensor from a Threonine to a Serine (SynCut3-S19) also leads to an earlier initial rise in net phosphorylation[44]. This is consistent with PP2A-B55 dephosphorylating pSerine, on average, less efficiently than pThreonine residues (Supplementary Fig. 6e), while CDK prefers to phosphorylate pSerines. These data show that PP2A-B55 plays a role in restricting this substrate from being net phosphorylated earlier during the cell cycle and contributing to the

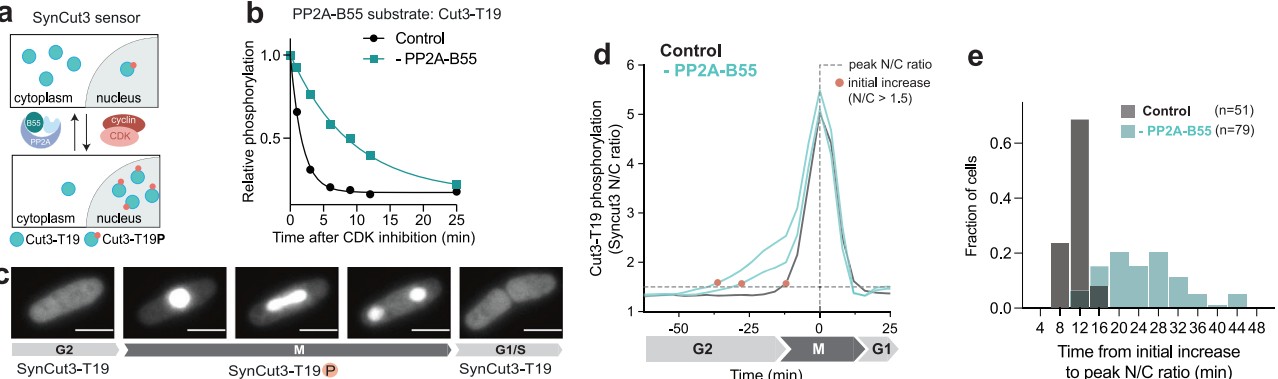

**Fig. 5 | In the absence of PP2A-B55 the CDK substrate site Cut3-T19 is phosphorylated earlier and less rapidly. a** Schematic of SynCut3-mCherry sensor. **b** Relative phosphorylation level of Cut3-T19 upon CDK-inhibition in the presence (black) and absence (turquoise) of PP2A-B55. Curves are a one-phase exponential decay fitted to the relative phosphorylation level determined by phosphoproteomics. **c** Representative fluorescent images of SynCut3-mCherry throughout a cell cycle within a single fission yeast cell, nuclear import of the sensor and binucleation can be seen in image 2-4. Error bars correspond to 5 μm. Representative of *n* > 50 cells, in *n* = 3 biological repeats. **d** Representative single-cell traces of SynCut3-mC N/C ratio in the presence (black) and absence (turquoise) of PP2A-B55, as determined by live-cell fluorescent imaging. Traces are aligned at their peak N/C ratio. The initial increase in N/C ratio above 1.5 is indicated by orange dots. **c, d** Representative of *n* > 50 cells, in *n* = 3 biological repeats, full data shown in Supplementary Fig. 11b. **e** Histogram of time between initial increase (N/C ratio >1.5) and peak N/C ratio in the presence (*n* = 52 cells) and absence of PP2A-B55 (*n* = 79 cells), representative of *n* = 3 biological repeats.

rapidness of the net phosphorylation increase at the G2/M transition.

## PP2A-B55 and CDC14 are rate-limiting for mitotic onset, independent of CDK activity

Given that all four phosphatases oppose G2/M CDK substrate sites, we investigated whether they play a rate-limiting role for the onset of mitosis. The phosphatases might act as 'thresholders', restricting the net phosphorylation of multiple CDK substrate sites during interphase, until CDK activity has reached a specific threshold. This could influence the timing of mitotic onset. In fission yeast, cell length at cell division can be used as a proxy for cell cycle timing (Fig. 6a, Supplementary Fig. 12a). As previously reported, in the absence of PP2A-B55 and CDC14, cells are accelerated into mitosis and thus divide at a shorter length (Fig. 6b)[46,47]. Cell length at division is not reduced in the absence of PP2A-B56[46], and in the absence of PP1, cells rapidly cease to grow in length, as PP1 activity is essential for polarised cell growth (Supplementary Movies 1&2)[48]. Thus, cell length cannot be used as a proxy of cell cycle stage in the absence of PP1.

PP2A-B55 and CDC14 are rate-limiting for mitotic onset, which could either be because the phosphatases dephosphorylate CDK substrates or they regulate CDK activity via the Tyr15 feedback loop[49,50] (Fig. 6c). To distinguish these two possibilities, we tested whether an advancement into mitosis is also seen in a CDK mutant background that is not subject to the Tyr15 feedback loop (CDK$^{AF}$ mutant)[22]. In the CDK$^{AF}$-mutant background, degradation of PP2A-B55 resulted in a 26% reduction in median cell length at division, while degradation of CDC14 led to a 13% reduction (Fig. 6d). This indicates that both phosphatases influence the timing of the G2/M transition in a manner that is independent of the regulation of CDK activity through the Tyr15 feedback loop. As a control, we also repeated this experiment with the tyrosine phosphatase Pyp1, which negatively affects CDK activity, by regulating the Tyr15 feedback loop via the Sty1 kinase[51,52]. Cells lacking Pyp1 are therefore accelerated into mitosis[46]. Consistent with this, degradation of Pyp1 in a wild-type CDK background reduced cell length at division (Supplementary Fig. 12b). However, degradation of Pyp1 in a CDK$^{AF}$-mutant background did not lead to a reduction in cell size (Supplementary Fig. 12c). This indicates that Pyp1's negative effect on the G2/M transition does not function independently of the Tyrosine15 feedback loop, unlike PP2A-B55 and CDC14.

Our data shows that PP2A-B55 and, to a lesser extent, CDC14, influence the timing of the G2/M transition independently of CDK

activity regulation via Tyrosine15. This suggests that in a CDK$^{AF}$ mutant background in the absence of PP2A-B55 and CDC14, critical CDK/PP2A-B55 and CDK/CDC14 substrates are net phosphorylated earlier in G2, thereby advancing cells into mitosis.

## Discussion

Our dataset provides a comprehensive resource of CDK-dependent phosphatase substrates in vivo. We identified substrates of four different phosphatases (PP2A-B55, PP2A-B56, PP1, and CDC14) implicated in the cell cycle, which together target 59% of all CDK substrate sites identified in vivo (Fig. 2b). Of these, over 90% were targeted by just one of the investigated phosphatases (Supplementary Fig. 5g), indicating that there is a division of labour between the phosphatases. Less than 5% of CDK substrate sites were targeted non-redundantly by two or three different phosphatases, such that the degradation of each of these phosphatases reduced the observed dephosphorylation rate of a specific site upon CDK inhibition. Substrates that are targeted by multiple phosphatases may act as hubs of phosphatase integration. One example for such a site is the replication initiation factor Sld3 (Supplementary Fig. 5h), whose dephosphorylation during S-phase is critical for proper origin firing in budding yeast[53]. We did not identify an opposing phosphatase for 41% of CDK substrate sites, possibly because these sites are targeted by other phosphatases, such as PP4 or PP6[7,54,55]. Alternatively, these sites may be redundantly targeted by multiple phosphatases, or degradation of the phosphatase of interest leads to compensatory kinase activity or broader network-level adaptations, so that the degradation of one phosphatase is entirely compensated by another, resulting in no discernible difference in the dephosphorylation rate upon depletion of any of the investigated phosphatases.

Our data show a clear division of labour between the investigated phosphatases in targeting CDK substrate sites, contrasting with work in budding yeast, which concluded that PP2A-B55, PP2A-B56 and CDC14 jointly target CDK substrate proteins and certain sites[13]. This study differed in several ways from ours. It compared progression from metaphase to G1 in the presence and absence of phosphatases using phosphoproteomics and electrophoretic mobility shifts. The latter is a protein-level analysis, which possibly underestimates substrate specificity, as our data show that sites on the same substrates are often phosphorylated by different phosphatases. In contrast, we identified phosphatase substrates by comparing dephosphorylation kinetics upon CDK inhibition after acute degradation of the

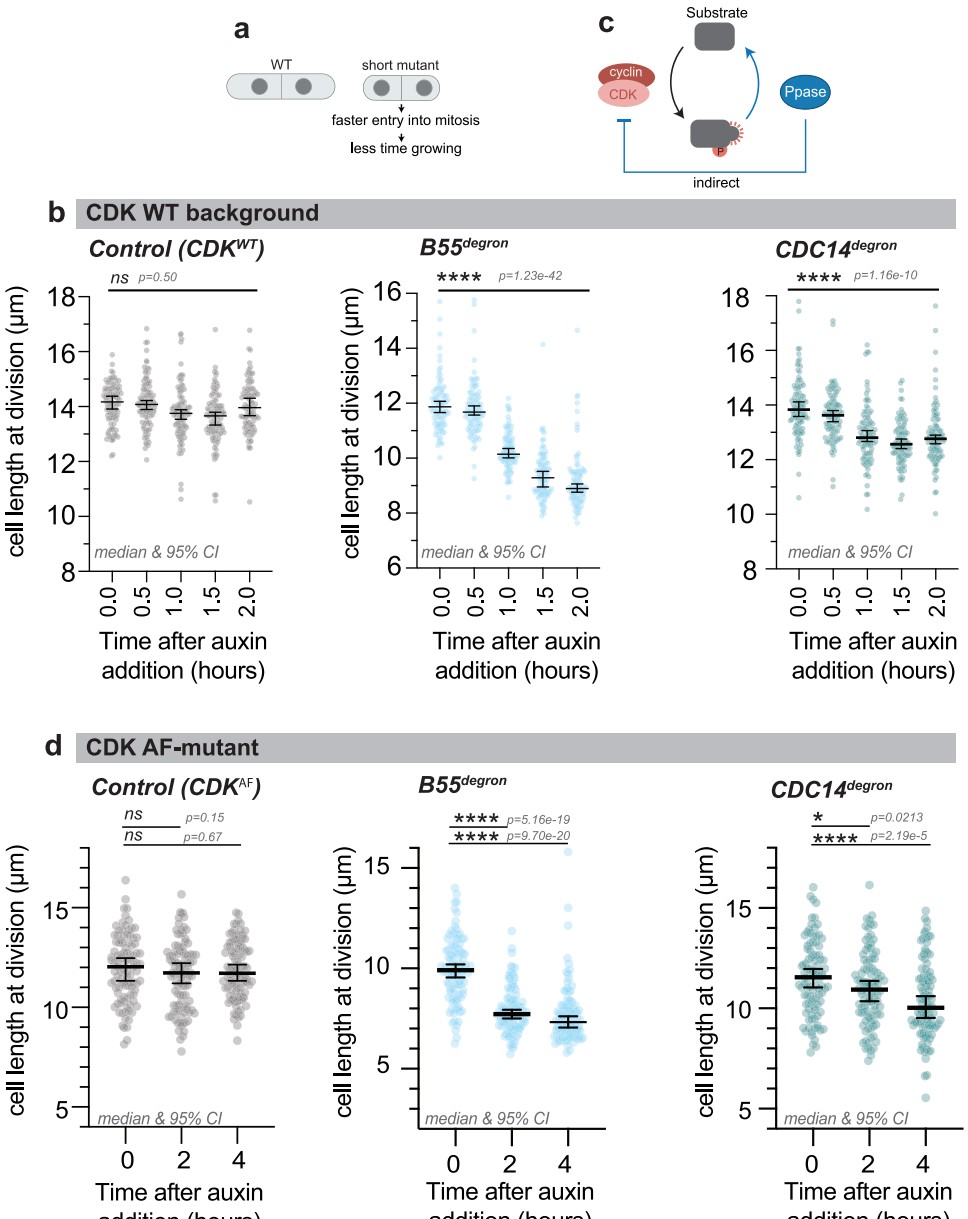

**Fig. 6 | Degradation of PP2A-B55 and CDC14 accelerates cells into mitosis, independently of CDK activity regulation via the Tyrosine15 feedback loop.** **a** Schematic of cell length at division of fission yeast cells. **b** Cell length at division of a *Control*, *PP2A-B55^degron^* and *Cdc14^degron^* strain at the indicated time after auxin addition (*n* = *100* cells per strain, representative of *n* = *3* biological repeats). **c** Schematic of the effect of phosphatase on CDK substrate phosphorylation, either directly by dephosphorylating CDK substrate sites or by affecting CDK activity. **d** Cell length at division of a *CDK^AF^*, *CDK^AF^ PP2A-B55^degron^* and *CDK^AF^ Cdc14^degron^* strain at the indicated time after auxin addition (*n* = *100* cells per strain, representative of *n* = *3* biological repeats). **b**, **d** Error bars represent median and 95% CI. Student's t-test (two-tailed, unpaired) was used to determine statistical significance.

phosphatases on a site-specific level. Thus, the discrepancy might be due to differences in the experimental approaches. It is also possible that at mitotic exit, phosphatase substrate specificity changes in comparison to interphase, with increased phosphatase activity at mitotic exit potentially reducing substrate specificities.

The exact time at which each CDK substrate site is net phosphorylated during the cell cycle will be determined by a combination of factors, including its sensitivity to CDK, presence of SLiMs and localisation (Fig. 7a). Our data have shown that the identity of the phosphatase that targets a CDK substrate site is an important determinant in this puzzle that sets the net phosphorylation timing. We have shown that CDK substrate sites dephosphorylated by CDC14 and PP2A-B56 are, on average, net phosphorylated earlier in G2, followed in order by CDK substrate sites dephosphorylated by PP1 and PP2A-B55

(Fig. 4a). The difference in net phosphorylation timing between the groups of phosphatase substrates may be due to differing catalytic efficiencies of the phosphatases, as increasing the turnover rate of a phosphosite can delay its net phosphorylation to a later point in the cell cycle. Alternatively, the differences in net phosphorylation time may be achieved by the activity of the different phosphatases being downregulated at different points during the cell cycle (Fig. 7b). CDK-opposing phosphatases are active during interphase, and their activities are downregulated to different extents during mitosis[7]. At mitotic exit, PP1 is reactivated first and then reactivates PP2A-B55[56,57]. A similar mechanism could potentially operate in reverse at the G2/M transition. Different phosphatases becoming downregulated at different points during the G2/M transition would allow their respective CDK substrate sites to be net phosphorylated at different times.

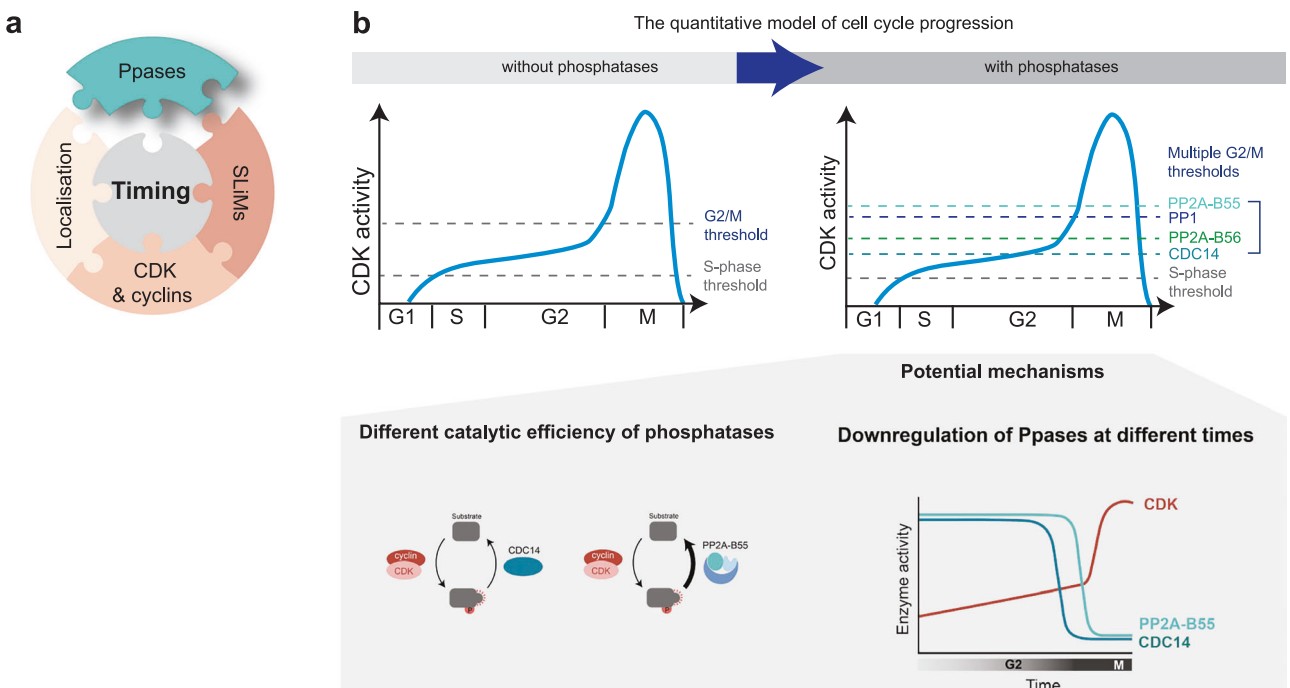

**Fig. 7 | The role of CDK-opposing phosphatases in regulating the timing of CDK substrate phosphorylation. a** Schematic of the different factors influencing the phosphorylation timing of CDK substrate sites. **b** Schematic of the effect of phosphatases on the threshold of the quantitative model of cell cycle progression and potential molecular mechanisms.

The quantitative model of the cell cycle proposes that progression through the cell cycle is ordered by an increase in net CDK activity, with cell cycle events happening at different net CDK activity thresholds[2,58] (Fig. 7b). Our data imply that rather than there being one mitotic threshold, there are multiple phosphorylation thresholds at the G2/M transition for different groups of CDK substrates targeted by different phosphatases (Fig. 7b). The division of labour between phosphatases can provide a finer time resolution at the G2/M transition, allowing for a more precisely ordered phosphorylation of hundreds of G2/M CDK substrates. This is important given the complexity of the transition from G2 into mitosis when hundreds of CDK substrates need to be phosphorylated in a precise order while CDK activity is rising rapidly[2,59,60]. It remains unclear which CDK substrates are essential for entry into mitosis. This is highlighted by our result that in the absence of PP2A-B55 and, to a lesser extent, CDC14, cells are accelerated into mitosis independently of CDK activity regulation via the Tyr15 feedback loop (Fig. 5d). This advancement is likely due to earlier net phosphorylation of a particular set of critical CDK substrates and could either result from PP2A-55 and CDC14 targeting different sites on the same critical proteins, or PP2A-B55 and CDC14 targeting different substrate proteins, which can independently advance cells into mitosis. We identified 20 proteins which contain both PP2A-B55 substrate sites and CDC14 substrate sites and might, therefore, be rate-limiting for mitotic entry. They include proteins involved in chromosome segregation (INCENP^Pic1, Survivin^Bir1), chromatin structure (Spt2, Cph1), transcription regulation (Bdp1) and nuclear-cytoplasmic transport (Nup60). Both phosphatases target other potentially rate-limiting substrates, which may independently advance cells into mitosis. Consistent with our data, deletions of CDC14 and the major catalytic subunit of PP2A have been shown to make cells more resistant to a decrease in global CDK activity[61], likely because in the absence of these phosphatases, CDK can net phosphorylate critical substrates earlier. Consistent with this, in budding yeast, PP2A-B55 delays the net phosphorylation of threonine sites during the cell cycle[19].

We conclude that the in vivo substrate specificity of the investigated phosphatases has regulatory consequences for the cell cycle, by affecting the net phosphorylation timing of CDK substrate sites. Given the high degree of conservation of cell cycle architecture and substrate specificity of the phosphatases, our conclusions are likely relevant for core cell cycle control in other eukaryotes.

## Methods

### *Schizosaccharomyces pombe* genetics and cell culture

Fission yeast media and growth conditions have been described previously[62]. Strains were constructed by transformation or mating as previously described[62] and checked for correct genotype by colony PCR (Supplementary Data 4). To generate the degron strains we used a TIR1^OSTIR1 strain and tagged the phosphatase of interest with a C-terminal sAID tag using standard transformation procedures (Supplementary Data 4)[24]. All experiments were performed with exponentially growing cultures in YE4S media (yeast extract with the supplements adenine, leucine, histidine, and uridine at 0.15 g/L) at 32 °C. Experiments with strains including the *cdc2(AFas)-cdc13 TIR1* background were performed in YE4S supplemented with 3% w/v galactose to prevent flocculation[63].

### Fluorescence microscopy

All microscopy experiments were performed using a Nikon Ti2 inverted microscope with Perfect Focus System, an Okolab environmental chamber and a Prime BSII sCMOS camera (Photometrics). Cells were imaged using an Olympus 100X objective lens with immersion oil (NA 1.45). The microscope was controlled with Micro-Manager v2.0 software (Open-imaging)[64]. For time-lapse imaging, cells were plated onto thin agarose pads, adapted from previous work in S. *cerevisiae*[65]. Cell masks were generated in Matlab and timelapse experiments analysed as previously described[66].

### Flow cytometry

To measure DNA content 1 mL of culture was pelleted; cells were fixed by addition of 70% v/v ice-cold ethanol and stored at 4 °C. Cells were

then pelleted (800 g, 3 min), washed with 50 mM sodium citrate and resuspended in 50 mM sodium citrate containing 0.1 mg/ml RNaseA (Sigma) for 18 h at 37 °C. 50 mM sodium citrate containing SYTOX Green (Thermo Fisher, final concentration of 1 μM) was added to samples, which were vortexed and sonicated for 10 s per sample (Soniprep 150 plus). At least 50,000 cells were analysed per sample on a BD LSRFortessa cell analyser. DNA area vs width (530/30-A/W) is plotted on a linear scale after gating for single cells in FlowJo (v10) based on Side-Scatter-Area vs Forward-Scatter-Area.

## Protein extraction

Protein extracts for proteomics and western blotting were prepared by adding ice-cold trichloroacetic acid (100%, w/v) to cell culture samples to a final volume of 10%. Cells were kept on ice for at least 30 min, pelleted at 3000 g, and washed in ice-cold acetone, before dry pellets were stored at −80 °C. Pellets were washed and resuspended in lysis buffer (8 M urea, 50 mM ammonium bicarbonate, 5 mM EDTA (Ethylenediaminetetraacetic acid), 1 mM phenylmethylsulfonyl fluoride, protease inhibitor cocktail set III, and PhosSTOP phosphatase inhibitor cocktail). Acid wash glass beads (0.4 mm) were added to cell suspensions and beaten (using FastPrep120) to break cells. Cell debris was pelleted out and supernatant recovered as a protein extract (stored at −80 °C).

## Western blotting

Protein detection by western blotting was performed for AID-tag using a 1:1000 dilution of anti-mAID (mouse monoclonal antibody, MBL, M214-3), blocked with 5% milk in TBS-T, for TPxK-phosphorylation using a 1:1000 dilution of anti-pTPxK antibody (rabbit monoclonal, Cell Signalling Technology, D9V5N), blocked in 5% BSA in TBS-T, for Cdc2-pTyr15 using a 1:500 dilution of phospho-Cdc2-Tyr15 antibody (rabbit polyclonal, Cat#9111; RRID: AB_331460), blocked in 5% BSA in TBS-T, for Cut12-p75 using a 1:200 dilution of phospho-Cut12-p75 antibody (Hagan lab[67]), blocked in 5% milk in TBS-T, for Cut12 using a 1:2000 dilution of Cut12 antibody (Hagan lab[67]), blocked in 5% milk in TBS-T. for Spo15-pS121 using a 1:200 dilution of phospho-Spo15-S121 antibody, blocked in 5% milk in TBS-T, for Spo15 using a 1:200 dilution of Spo15 antibody, blocked in 5% milk in TBS-T. The Spo15-pS121 and Spo15 antibodies were generated and purified by an external company (Covalab). Antisera were incubated with a non-phosphorylated peptide to isolate the Spo15 control antibody, and the remaining antisera were incubated with a phosphorylated peptide to isolate the Spo15-pS121 antibody. The secondary antibodies used in this study were: 1:5000 mouse IgG HRP-linked whole antibody from sheep (GE Healthcare Cat# NA931, RRID:AB_772210); 1:5000 rabbit IgG HRP-linked whole antibody from donkey (GE healthcare, Cat# NA934, RRID:AB_772206), 1:10,000 sheep IgG HRP-linked whole antibody (Noves, Cat# A16041). Signal was detected using SuperSignal West Femto Maximum Sensitivity Substrate (34095, Life Technologies) and imaged on an Amersham Imager 600.

## Co-immunoprecipitation mass spectrometry

Protein extracts for immunoprecipitation were prepared via cryomilling. Cells were collected by centrifugation (3 min, 2000× g), resuspended in buffer A (50 mM Tris-HCl pH 8.0, 50 mM KCl, 2 mM MgCl$_2$, 0.2% TritonX-100, 1X cOmplete EDTAfree protease inhibitor cocktail (Roche), 1X phosSTOP phosphatase inhibitor cocktail (Roche)) and frozen in small pellets by drop-wise addition into liquid nitrogen. Cells were ground in a CertiPrep 6850 Freezer/Mill for 3×2 min cycles. The powder was stored at −80 °C and aliquots were resuspended in ice-cold buffer A, briefly sonicated using a Bioruptor (20 s) and centrifuged (5 min, 16,000 g, 4 °C). The supernatant was incubated with Protein G dynaBeads (ThermoFisher), cross-linked to anti-GFP antibodies (Roche) at 4 °C for 40 min. Beads were washed 3 times in buffer A and resuspended in elution buffer (10% SDS in 100

mMTEAB). Samples were digested using S-trap, and data were acquired using the Evosep system coupled to TIMS TOF Ultra using DIA PASEF. The Evosep system was fitted with an Ion Optics Aurora Elite column. Whisper Zoom 40SPD method was used for chromatography. Data were processed using Spectronaut19 using standard settings and an *S. pombe* Uniprot database.

## Tandem mass tag proteomics

Each protein sample (300 μg) was reduced with 5 mM dithiothreitol (DTT) for 25 min at 56 °C, alkylated with 10 mM iodoacetamide (30 min, room temperature, dark) and then quenched with 7.5 M DTT. Samples were digested using S-Traps (Protifi) spin columns, with the variation that 50 mM HEPES (pH 8.5) was used in place of ammonium bicarbonate. The digested samples were labelled with the TMTpro 16plex Isobaric Label Reagent Set (Thermo Fisher) according to the manufacturer's instructions. After labelling and mixing, multiplexed samples were desalted using a Thermo desalting column. Phosphopeptide enrichment was performed using sequential enrichment from metal oxide affinity chromatography (SMOAC, Thermo Fisher). Initial enrichment was done using the HighSelect TiO2 Phosphopeptide Enrichment kit followed by the HighSelect Fe-NTA Phosphopeptide Enrichment kit (both from Thermo Scientific). Phosphopeptides and non-bound flow-through fractions were desalted and fractionated using the High-pH Reversed-Phase Peptide Fractionation kit (Pierce) and analysed on Orbitrap Eclipse Tribrid mass spectrometer (Thermo Fisher) coupled to an UltiMate 3000 HPLC system for online liquid chromatography separation. Each run consisted of a 3-hour gradient elution from a 75 μm × 50 cm C18 column operated in Data-dependent acquisition mode.

## Mass spectrometry data analysis

MaxQuant (version 1.6.14.0, and 2.4.7.0) was used to search data against an *S. pombe* proteome FASTA file, extracted from PomBase and amended to include common contaminants. Default MaxQuant parameters were used apart from adding phospho(STY) as a variable modification for phosphopeptide-enriched samples. Phospho(STY).txt (phosphopeptides) and ProteinGroup.txt (peptides) MaxQuant outputs were imported into Perseus (version 1.6.14.0) for further analysis. Data was filtered for valid values (100%), localisation probability (> 0.75) and median normalised. The same phosphosite with a different phosphorylation multiplicity was treated as a separate phosphorylation event. All further analyses, including curve fitting and plotting of heatmaps, were done using Matlab (R2024b) and RStudio. Phosphatase substrates (proteins and sites) are listed in Supplementary Data 1 (CDK-dependent phosphatase substrates) and 2 (CDK-independent phosphatase substrates). Enrichment logos were made using iceLogo[68], GO enrichment analysis was conducted using ShinyGO[69] and the disordered regions of the *S. pombe* proteome were searched for SLiMs SLiMSearch4[70]. The phosphoproteomic timecourse analysed in Fig. 4 and Supplementary Fig. 6 was previously described[2]. In brief, a Cdc13-Cdc2-as strain in which Cig1, Cig2, Puc1 had been deleted (ΔCCP), was grown in EMM4S + SILAC (stable isotope labelling with amino acids in cell culture) media at 32 °C. Cells were arrested at G2 using 1 μM 1-NmPP1 and released into mitosis by washing into 1-NmPP1-free media[2].

## Phosphatase substrate identification

Phosphoproteomic data of Control and -Ppase condition normalised to their respective timepoint 0 (mean of two technical replicates was used for t0) were used to identify CDK substrate sites and CDK-dependent phosphatase substrates (Supplementary Fig. 1c). CDK substrate sites were defined as sites that fitted the following criteria. i) Phosphorylation of sites decreased by at least 50% within 12 min after CDK inhibition. ii) An exponential decay function was fitted to these sites using the non-linear least-squares method in Matlab, calculating

the dephosphorylation rate constant and half-life. CDK substrates were defined as sites that fitted an exponential decay with R-squared > 0.85. To identify Ppase substrates, the integral was calculated for all CDK substrates between the control and -Ppase condition (t0-t25). A site was defined as a phosphatase substrate if i) the site fitted an exponential decay with R-squared > 0.7 and ii) the difference between the integrals (-Ppase - Control) was >5. Sites with an integral difference <5 and >1 were defined as phosphatase substrates if the confidence intervals (0.8) of the fitted k-values of the Control and -Ppase condition did not overlap. For CDK-independent phosphatase substrates, a linear regression was fitted to the phosphoproteomic data (not normalised to t0) of the control and -Ppase condition using RStudio. Sites were defined as CDK-independent if the slope was >−0.05 and <0.05 and the RMSE < 0.4. A student's t-test (two-tailed, corrected for multiple testing using the Benjamini-Hochberg method) was used to determine the statistical difference between the mean phosphorylation level of the Control and -Ppase condition (Supplementary Fig. 5a).

## Statistical analysis
All statistical tests were conducted using RStudio 2024b or GraphPad Prism 8. The central point for data given is either the mean value, with whiskers indicating 95% CI, or the median value, with whiskers indicating 95% CI or IQR, unless otherwise stated.

## Reporting summary
Further information on research design is available in the Nature Portfolio Reporting Summary linked to this article.

## Data availability
All mass spectrometry data generated for this study have been deposited to the ProteomeXchange Consortium via the PRIDE partner repository with dataset identifier PXD060298, and sample names are listed in Supplementary Data 5. Mass spectrometry data for Fig. 4 and Supplementary Fig. 10 can be found on PRIDE with dataset identifier PXD003598[2]. Source data are provided with this paper.

## Code availability
Previously described custom scripts used to analyse fluorescent time-lapse data can be found at https://github.com/nkapadia27/Spatiotemporal-Orchestration-of-Mitosis[66].

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

## Acknowledgements

We thank J.Curran, J. Greenwood, B. Whyte, S. Willich and T. Hammond for their comments on the manuscript, J. Curran for help with plasmid construction, T. Carr and A. Watson for *S. pombe* strains with the OsTIR1(F74A)-NLS construct, L.Du for the auxin-tagging plasmid and I. Hagan for sharing antibodies. This work was supported by the Francis Crick Institute, which receives its core funding from Cancer Research UK (CC2003), the UK Medical Research Council (CC2003), and the Wellcome Trust (CC2003). In addition, this work was supported by the Wellcome Trust Grant to P.N. (grant number 214183), The Lord Leonard and Lady Estelle Wolfson Foundation, Woosnam Foundation and The

Breast Cancer Research Foundation (BCRF-23-117). T.U.Z. received funding from the Boehringer Ingelheim Fonds. For the purpose of Open Access, the author has applied a CC BY public copyright licence to any Author Accepted Manuscript version arising from this submission.

## Author contributions
T.U.Z. and P.N. initiated the study. T.U.Z. designed and performed all experiments. T.A. and T.U.Z. performed mass spectrometry experiments. E.L.R. developed the Spo15 protein and phospho-antibodies. T.U.Z. performed data analysis. T.U.Z. and P.N. wrote the manuscript with input from all authors.

## Funding

## Competing interests
The authors declare no competing interests.
