## [Transparent Peer Review file · Nature Communications]

Phosphatase specificity influences phosphorylation timing of CDK substrates during the cell cycle

Corresponding Author: Dr Theresa Zeisner

Version 0:

Reviewer comments:

Reviewer #1

(Remarks to the Author)

In this paper, the authors used a degron approach to systematically investigate the contribution of protein phosphatases to CDK substrate phosphorylation timing in the cell cycle of fission yeast, specifically in G2/M transition. As the authors use an unbiased mass spectrometry approach, they identify numerous phosphorylation sites that are regulated by a specific phosphatase. By doing so, the authors identified CDK substrates that are (largely uniquely) regulated by the 4 investigated phosphatases (CDC14, PP1, PP2A-B55 and PP2A-B56). They describe that CDK phosphorylation events are counteracted early in G2 by CDC14 and PP2A-B56, followed later in G2 by PP1 and then PP2A-B55. The analysis is also expanded to non-CDK kinases and through bioinformatic approaches signature “dephosphorylation” motifs are generated for the 4 phosphatases.

In general, the experimental approaches are of high quality as well as the data that the authors have compiled and analysed. However, a major limitation is that the conclusions are largely in line with what is known from studies in other model systems and expand on similar studies performed in mitotic exit (e.g., F. Uhlmann lab in budding yeast). Thus, the study does not add much new insight and clearly the complexity of how phosphatases select substrates is not addressed with this work. The study could be strengthened if the authors invested more in zooming in on specific substrates and show how their specific dephosphorylation contributes to cellular functions.

Major comments:

- 1) The authors should expand on the known crosstalk between phosphatases and between phosphatases and CDK.
- 2) It would be interesting to see whether CDK substrates targeted by more than one phosphatase are known hubs of phosphatase activity integration or are more excitingly maybe even novel points of phosphatase interaction, dictating timing of the cell cycle.
- 3) Does depletion of phosphatases affect phosphorylation and activity of CDKs themselves?
- 4) Similarly, what about phosphorylation of phosphatases?
- 5) How does inhibition of these phosphatases affect timing of cell cycle transitions under unperturbed conditions?
- 6) What about known substrates versus novel substrates? The dataset of known substrates seems to be rather minimal as compared to total amount of affected phosphorylation sites, so this should be highlighted better.
- 7) The last part of the results focusing on phosphatase effects in Y15 mutated cells seems detached to the rest of the story.

Specific and minor comments:

- 1) Result section 1 starting at line 77: the authors make a claim on the timing regulation of total CDK activity, while only *cdc2* is inhibited. We have no problem if the authors would for example state something down the line of ‘inhibition of *cdc2* activity, hereafter referred to as CDK activity’. However, the general claims on total CDK activity in the current shape should be removed in their entirety and geared towards *cdc2*.
- 2) As *cdc2* can also affect G1/S transition can this confound the data, claiming to look only at G2/M? Can the authors exclude that the effects on phosphorylation timing they observe are not taking place in G1/S? Can the authors provide information on timing of transition from G2/M after *cdc2* inhibition?
- 3) Same results section 1: It is unclear to us how these degron cells were generated. Can the authors provide a better

description in the materials and methods? provide a reference to the provided table with all strains for clarity.

- 4) The schematic in fig1a should be corrected to reflect the actual experimental flow, where *cdc2-as* inhibition takes place first, and then degradation of the phosphatase is achieved, followed by release (this is how we understand it from the extended data 1b panel and as this is more complete, we assume this is the correct version).
- 5) Line 133: 'countering the notion that phosphatases are unspecific'. This claim lies long behind us and should be rephrased.
- 6) Section starting at line 202: could you perform this analysis for CDK-substrates alone as well?
- 7) Line 220: instead of 'data show', this should be 'data further confirm', since this is not new.
- 8) Section starting at line 222: there is a lot of switching between 'median phosphorylation', 'net phosphorylation' and 'relative phosphorylation'. Are they pointing to the same? If yes, choose one. In any case, they should be defined better, for example, how is relative phosphorylation, used in graphs, calculated?
- 9) Line 232: why data cropped at 40 min? Would be informative to see the entire profile.
- 10) Line 239: we understand the authors use AUC as a proxy for timing of onset, but this should be made more clear in the text and not just in the figure legend. Also, can the distribution of relative phosphorylation be shown instead of a single line?
- 11) Line 244: S-phase substrates. Where is this data shown? This could be effects of *cdc2-as* inhibition on G1/S transition.
- 12) Line 268: please swap legend order of CDC14 and B55 for consistency
- 13) Line 271: Please revisit this claim. B55 substrates are more affected in nucleus and at SPB as compared to cytoplasm, as is evident from extended data fig 6b.
- 14) Also, please describe abbreviations (e.g. SPB) – would be good if authors could double check this throughout text in general.
- 15) Section starting line 279: we appreciate this section contributes to the relevance of *cut3* as a novel b55 substrate. An important shortcoming here is that for b55 the auxin inducible system was used, and for the other phosphatases stable deletion lines were used. Can the authors elaborate on the effect on timing of fraction of cells in peak N/C *cut3-mcherry* as seen for the other phosphatases.
- 16) Can to authors identify a direct interaction between B55 and *cut3*?
- 17) Section starting at line 336: Please rephrase this section to be clearer.
As it becomes clear from this data, b55 degradation has a stronger effect than *cdc14*, and therefore accelerates entry into mitosis faster than B55. If B55 acts later than *cdc14*, as the authors state, but the effect on B55 is larger,
- 18) Line 366: Secondary effects on phosphorylation should be considered.
- 19) Line 430: this should be rephrased. Net phosphorylation is affected, not just phosphorylation by cdk.

Reviewer #2

(Remarks to the Author)

The cell cycle is tightly regulated by the coordinated actions of cyclin-dependent kinases and their counteracting phosphatases. While CDK-cyclin complexes phosphorylate numerous substrates in a cell cycle-dependent manner, phosphatases play a crucial role in reversing these modifications to ensure proper cell cycle progression. Despite extensive research on CDK-cyclin regulation, the mechanisms by which phosphatases fine-tune the timing and specificity of CDK-mediated phosphorylation remain poorly understood.

To address this critical gap in knowledge, the authors employed *Schizosaccharomyces pombe* as a model system, using phospho-proteomics to systematically analyze phosphorylation dynamics following perturbation of the kinase Cdc2 and four phosphatases: PP2A-B55, PP2A-B56, PP1, and CDC14. Their study successfully identified a broad spectrum of phosphatase-specific substrates, demonstrating that each phosphatase targets distinct pools of Cdc2-phosphorylated proteins. Furthermore, the authors characterized the sequence features of phosphatase-targeted phosphosites, providing mechanistic insights into substrate recognition. Notably, their findings suggest that phosphatases serve as an additional regulatory layer, modulating CDK phosphorylation thresholds required for mitotic entry.

The study is technically rigorous, with well-executed experiments and appropriate data interpretation. However, the following revisions could further strengthen the manuscript.

- 1) To confirm the robustness of the phospho-proteomic data, Western blot analysis of known Cdc2 and phosphatase substrates should be included. This would provide validation of phosphorylation changes observed in the mass spectrometry experiments.
- 2) The study identifies numerous new phosphatase substrates. However, further biochemical validation (e.g., in vitro dephosphorylation assays or mutagenesis of phosphosites) would strengthen the biological relevance of these findings. At least one representative new substrate for each phosphatase should be experimentally verified.
- 3) To reinforce the claim of phosphatase-substrate specificity, biochemical assays (e.g., phosphatase inhibition or substrate competition experiments) should be performed on selected substrates.

Reviewer #3

(Remarks to the Author)

Zeisner et al engineer *S. pombe* lines with ATP-analog alleles of Cdc2 and AID-tagged alleles of either B55, B56, PP1, and CDC14. They perform time course phosphoproteomics analysis upon inhibition of Cdc2/CDK and fit their data to an exponential decay curve to identify Cdc2/CDK substrate phosphorylation sites and their opposing phosphatase. They

identify kinase–phosphatase interactions and find little site overlap due to differences in phosphatase amino acid preferences for dephosphorylation sites and SLiM recruitment. Furthermore, they temporally order Cdc2/CDK opposing phosphatase activities and determine that each sets a distinct phosphorylation threshold coordinating specific Cdc2/CDK-driven events during mitotic entry. This is an important addition to the regulatory model of mitotic entry, as it more explicitly defines the contribution of ordered phosphatase inactivation to the increase in phosphorylation at the G2/M transition. Overall, this is a comprehensive and well-carried-out analysis of the Cdc2/CDK opposing phosphatases in *S. pombe* that greatly adds to our understanding of these phosphatases and their contribution to cell cycle regulation. The authors investigate B55, B56, PP1, and Cdc14 under the same experimental conditions, allowing for a robust comparison of datasets to determine phosphatase specificity based on the dephosphorylation of CDK-dependent and -independent sites. Results obtained from these studies reproduce observations made in human cells, demonstrating the conservation of the mechanisms underlying phosphatase specificity and substrate recognition. The time course analysis of dephosphorylation increases the power of the analysis, allowing them to measure the impact on the timing of Cdc2/CDK phosphorylation. Extended data 1. D/E: Why did the authors observe such highly variable numbers of phosphorylation sites in each of the four phosphoproteomic experiments? 1F: How many sites did the authors detect in the proline-directed versus non-proline-directed category?

Extended data 2B. The differences in pTPxK levels are not noticeable. Quantification and normalization to ponceau staining would help to visualize these differences.

The authors state that: “they classified CDK substrate sites as phosphatase targets if their dephosphorylation was substantially slower in the absence of one of the phosphatases”. Here, clearly stated cutoffs should be used. Was any statistical test performed?

In addition to their scheme, the authors should include examples of known B56 or Cdc14 CDK-independent substrates in Figure 3a.

A bar graph directly comparing the number of CDK-dependent and -independent substrates identified across all experiments for phosphatase should be included in the main text Figure 3. The data shown in extended data Figure 5c for Cdc2/CDK-independent substrates seems high for PP2A-B55 and PP1. Do the authors have an explanation for this? Representing these sites as a weblogo, in addition to the icelogo, and performing a motif enrichment analysis, would provide additional information on the type of sites dephosphorylated by each of the phosphatases.

The authors' analysis would be more impactful if they could perform an additional phosphoproteomics time course analysis for a different kinase for which they predict a specific phosphatase opposition course analysis.

The timing analysis is very interesting. The authors infer phosphatase activity based on when their substrates are phosphorylated by Cdc2/CDK in G2/M. However, the inferred effect on phosphatase activity is never stated. It would help the reader if the authors would do so and, in addition to their diagram of Cdc2/CDK activity thresholds, provide a model for phosphatase activity levels in G2/M.

Version 1:

Reviewer comments:

Reviewer #1

(Remarks to the Author)

The authors have addressed concerns and recommendations from all reviewers and I support publication.

Reviewer #2

(Remarks to the Author)

The authors have addressed my concerns.

Reviewer #3

(Remarks to the Author)

The authors have addressed my concerns and suggestions. This is a great manuscript, I am looking forward to seeing it in press.

General response to all reviewers:

We thank the editor and the reviewers for their comments and the opportunity to revise and improve our work. In response, we have performed additional experiments and data analyses as well as corrected and clarified the text to improve our manuscript.

Below is an overview of the main changes:

- New western blots using phosphospecific antibodies to confirm phosphatase substrates identified in our phosphoproteomic dataset (Supplementary Fig. 4)
- New experiment showing interplay between phosphatase and CDK activity using western blots and phosphoproteomic data (Supplementary Fig. 7)
- New flow cytometry experiments showing cell cycle profiles in the absence of the investigated phosphatases (Supplementary Fig. 3a-b)
- New live-cell imaging data highlighting phenotypes upon PP1 degradation (Supplementary Movie 1&2)
- New co-immunoprecipitation experiments showing a direct interaction between Cut3 and PP2A (Supplementary Fig. 11a)
- New experiments showing the phosphorylation of synCut3-mCherry in a PP2A-B55 deletion strain (Supplementary Fig. 11c)

We have rephrased the text to highlight the importance and novelty of our *in vivo* work and the regulatory consequences this has for CDK substrate phosphorylation. Additionally, we have added a section on the interplay between kinases and phosphatases as well as describing example substrates and their biological relevance, as requested by Reviewer 1.

We outline our response (in blue) to each reviewer's comments (in black) below.

Reviewer #1 (Remarks to the Author)

In this paper, the authors used a degron approach to systematically investigate the contribution of protein phosphatases to CDK substrate phosphorylation timing in the cell cycle of fission yeast, specifically in G2/M transition. As the authors use an unbiased mass spectrometry approach, they identify numerous phosphorylation sites that are regulated by a specific phosphatase. By doing so, the authors identified CDK substrates that are (largely uniquely) regulated by the 4 investigated phosphatases (CDC14, PP1, PP2A-B55 and PP2A-B56). They describe that CDK phosphorylation events are counteracted early in G2 by CDC14 and PP2A-B56, followed later in G2 by PP1 and then PP2A-B55. The analysis is also expanded to non-CDK kinases and through bioinformatic approaches signature “dephosphorylation” motifs are generated for the 4 phosphatases.

In general, the experimental approaches are of high quality as well as the data that the authors have compiled and analysed. However, a major limitation is that the conclusions are largely in line with what is known from studies in other model systems and expand on similar studies performed in mitotic exit (e.g., F. Uhlmann lab in budding yeast). Thus, the study does not add much new insight and clearly the complexity of how phosphatases select substrates is not addressed with this work.

While phosphatase substrate specificity has been characterised biochemically, we show for the first time that a division of labour between phosphatases targeting CDK substrate sites *in vivo* affects net phosphorylation timing during the cell cycle. Previous biochemical studies revealed that PP2A-B55, PP2A-B56, PP1 and CDC14 utilise distinct substrate targeting mechanisms and differ in their amino acid preference and SliMs (Gray *et al*, 2003; Cundell *et al*, 2016; Kruse *et al*, 2020; Hoermann *et al*, 2020). However, it was still an open question whether *in vivo* different phosphatases target overlapping substrate groups and, in particular, what regulatory consequences this may have for the cell cycle. We show here for the first time that four phosphatases target mostly distinct groups of CDK substrates *in vivo*, and that this has consequences for phosphorylation timing within the cell cycle.

Work in budding yeast by the Uhlmann lab has previously concluded that “multiple Cdk-counteracting phosphatases show an overlapping substrate spectrum”(Touati *et al*, 2019), and their recent paper further concludes that the impact of phosphatases does not predict the timing of CDK substrate phosphorylation (Takacs *et al*, 2025). Our data present a contrasting picture. We show that the PP2A-B55, PP2A-B56, CDC14 and PP1 target distinct subgroups of CDK substrates. We furthermore make the novel observation that CDK substrates targeted by different phosphatases are net phosphorylated at different points during the cell cycle. Thus, our data provide new insight showing that the identity of a phosphatase that targets a site is an important factor in setting its net phosphorylation timing.

The different conclusions from our paper and the work from the Uhlmann lab are likely due to differences in experimental approach.

- In contrast to us, the Uhlmann lab compared a phosphoproteomic timecourse from metaphase to G1 in the presence and absence of Cdc55 (PP2A-B55), Rts1 (PP2A-B56) and Cdc14. Due to the use of deletion strains (or slow-acting degradation for Cdc14) rather than rapid depletion and the use of a long timecourse, their experiment likely identified downstream targets rather than just direct substrates. In contrast to this, we identified phosphatase substrates by comparing dephosphorylation kinetics within minutes after CDK inhibition and acute phosphatase degradation.
- Their claim that phosphatases target overlapping groups of substrates is partly based on electrophoretic mobility shifts of the CDK substrates Ask1, Orc6, Cbk1 in the presence and

absence of the phosphatases (Touati *et. al.*, 2019, Figure 2). This is a protein-wide analysis, but our data show that different phosphosites on the same protein are often targeted by different phosphatases (Figure 2c). Protein-based analyses, which lack phosphosite resolution, are therefore likely to be confounded by the action of multiple phosphatases.

Their claim in their most recent paper, that “phosphatases do not predict the timing of CDK substrate phosphorylation” is based on a comparison between dephosphorylation rate and phosphorylation timing of different CDK substrates, but, unlike our study, does not include any experiments that are done in the absence of these phosphatases. Furthermore, all phosphosites on a given protein were averaged to determine phosphorylation timing in their data analysis. We have shown that sites on the same protein are often opposed by different phosphatases. Averaging the phosphorylation behaviour of multiple phosphosites therefore risks inappropriately averaging potential differences in phosphorylation timing and concluding that there are no such differences.

We are confident that our paper provides novel insights into how phosphatase substrate specificity can shape the timing of CDK substrate phosphorylation *in vivo*. We have now made text changes throughout the manuscript to highlight this point and the novelty of our findings.

The study could be strengthened if the authors invested more in zooming in on specific substrates and show how their specific dephosphorylation contributes to cellular functions.

We have now highlighted multiple specific substrate examples throughout the text, detailed below. We show that in the absence of PP2A-B55 Cut3-T19 becomes phosphorylated earlier during the cell cycle and less rapidly (Figure 5d). A similar trend is seen when Cut3-T19 is mutated to a Serine, making it a worse PP2A-B55 substrate (line 336-340). This highlights that dephosphorylation by a phosphatase plays an important role in setting the timing as well as rapidness speed of net phosphorylation and thus nuclear import of Condensin (line 340-342). We have also added new data showing that PP1 degradation halts polarised cell growth (Supplementary Movie 1&2, line 120-123), consistent with PP1 being crucial for polarised cell growth in *S. pombe* (Kokkoris *et al.*, 2014). The polarity factors Tea4 and PP1 are brought to the cell tips by interphase microtubules, where they instruct cell growth via Cdc42 activation and the actin network. We identified a number of PP1 substrate sites on the polarity factor Tea4 and the actin-binding protein Wsp1 (Supplementary Figure 2d).

We also show that cells are accelerated into mitosis in the absence of PP2A-B55 and CDC14, independently of CDK activity regulation (Figure 6d). This advancement is likely due to earlier net phosphorylation of a particular set of critical CDK substrates and could either result from PP2A-55 and CDC14 targeting different sites on the same critical proteins, or PP2A-B55 and CDC14 targeting different substrate proteins, which can independently advance cells into mitosis. We identified 20 proteins which contain both PP2A-B55 substrate sites and CDC14 substrate sites and might, therefore, be rate-limiting for mitotic entry. They include proteins involved in chromosome segregation (INCENP^{Pic1}, Survivin^{Bir1}), chromatin structure (Spt2, Cph1), transcription regulation (Bdp1) and nuclear-cytoplasmic transport (Nup60). Both phosphatases target other potentially rate-limiting substrates, which may independently advance cells into mitosis. Both phosphatases target other potentially rate-limiting substrates, which may independently advance cells into mitosis. For example, we show that PP2A-B55 targets the Condensin subunit Cut3 (Supplementary Table 1, text line 447-452).

Major comments:

1) The authors should expand on the known crosstalk between phosphatases and between phosphatases and CDK.

We have added a section to the results highlighting cross-talk between CDK and phosphatase activity (line 174-183) and a schematic and example substrate sites in Supplementary Fig. 7c-d. We have also probed Cdc2-Tyr15 phosphorylation after phosphatase degradation using Western blotting (see also response to point 3&4).

2) It would be interesting to see whether CDK substrates targeted by more than one phosphatase are known hubs of phosphatase activity integration or are more excitingly maybe even novel points of phosphatase interaction, dictating timing of the cell cycle.

We thank the reviewer for this comment. We have now added examples of proteins, including the replication initiation factor Sld3 and the Cdc25 phosphatase, as examples of sites that are jointly targeted by multiple phosphatases (Supplementary Fig. 5h,i). We have also highlighted the point that these sites may be hubs of phosphatase integration into the discussion (line 387-390).

3) Does depletion of phosphatases affect phosphorylation and activity of CDKs themselves?

We have now added data showing that in the absence of PP2A-B55, Cdc2-Tyr15 phosphorylation levels are substantially decreased, indicating higher CDK activity. In the absence of PP2A-B56, CDC14 and PP1 Tyr15 levels are reduced, but to a lesser extent (new Supplementary Fig. 7a, line 175-178). We additionally performed new Western blots following Tyr15-phosphorylation after degrading the individual phosphatases in a G2 arrest. In these conditions, a decrease in Tyr15 phosphorylation 30 min after PP2A-B55 degradation can be seen (new Supplementary Fig. 7b).

4) Similarly, what about phosphorylation of phosphatases?

We now discuss this point in a new section of the results, highlighting cross-talk between phosphatases (line 174-183, Supplementary Fig. 7c-d).

5) How does inhibition of these phosphatases affect timing of cell cycle transitions under unperturbed conditions?

We conducted new flow cytometry analysis in the presence and absence of the phosphatases of interest to investigate their effect under unperturbed conditions. Auxin was added to asynchronous cultures to degrade phosphatases and samples were taken over 3 hours for flow cytometry analysis. This highlighted that upon PP1 degradation, cells arrest in metaphase. These data have now been added to Supplementary Fig. 3. In addition, our data in Figure 6 show that under unperturbed conditions (asynchronous cell culture), degradation of PP2A-B55 or CDC14 leads to an advance into mitosis, as seen by the reduced length at septation (Figure 6). We added new data showing that degradation of PP1 in an asynchronous culture leads to a stop in length extension (new Supplementary Movie 1&2, line 120-123). Live cell imaging of PP1-degredon strains with the synCut3-mCherry sensor, used as a mitotic marker, showed that cells become arrested in mitosis and show severe chromosome segregation defects (Supplementary Fig. 2c).

6) What about known substrates versus novel substrates? The dataset of known substrates seems to be rather minimal as compared to total amount of affected phosphorylation sites, so this should be highlighted better.

We list examples of known phosphatase substrates in Supplementary Table 1, to highlight that our phosphoproteomic experiments also identified already known phosphatase substrates, which were identified in a range of model organisms via different experimental approaches. There are only two studies which identified phosphatase substrates in *S. pombe* on a proteome-wide scale; however, both of these studies identify phosphatase substrates on a protein, not a site-specific level. Chen *et. al.* identified 73 Clp1/Cdc14 substrates using a

substrate trapping mutant approach. We identified CDK substrate sites on 43 of these proteins, of which around 60% contain at least one Cdc14 substrate site. Bernal *et. al.*, identified 27 potential PP2A substrates using 2D-DIGE analysis of PP2A complex mutants. We identified CDK sites on one of these proteins (hexokinase 1), which are dephosphorylated by PP2A-B56. These substrates have now been added to Supplementary Table 1.

7) The last part of the results, focusing on phosphatase effects in Y15 mutated cells, seems detached from the rest of the story.

In the last part of the paper, we show that in the absence of PP2A-B55 or CDC14 cells are advanced into mitosis. Since these phosphatases affect CDK activity, we used a Cdc2-Y15-mutant to demonstrate that this effect is independent of their effect on CDK activity (Figure 6). While the experimental approach is somewhat different from the rest of the paper, we think that this section offers novel insight. It shows that both CDC14 and PP2A-B55 act as thresholds, indicating that in the absence of these phosphatases, their substrates can be phosphorylated earlier, advancing cells into mitosis. We have now added a schematic to Supplementary Fig. 12a, to explain the experimental design.

Specific and minor comments:

1) Result section 1 starting at line 77: the authors make a claim on the timing regulation of total CDK activity, while only *cdc2* is inhibited. We have no problem if the authors would for example state something down the line of 'inhibition of *cdc2* activity, hereafter referred to as CDK activity'. However, the general claims on total CDK activity in the current shape should be removed in their entirety and geared towards *cdc2*.

In fission yeast, there is only a single CDK, which is encoded by the *cdc2* gene. Thus, treating cells containing the analogue-sensitive *cdc2-as* allele with 10 μ M 1-NmPP1 inhibits all CDK activity. We have now changed the text to clarify this point (line 86-87).

2) As *cdc2* can also affect G1/S transition can this confound the data, claiming to look only at G2/M? Can the authors exclude that the effects on phosphorylation timing they observe are not taking place in G1/S?

The phosphoproteomic timecourse shown in Figure 4a includes both S-phase and mitotic CDK substrates, and we included all these CDK substrate sites in our analysis. Our data show that different S-phase substrates are targeted by different phosphatases, rather than all by the same phosphatase. This is stated in line 276-277. Thus, we do not claim that phosphatases or CDKs do not act at the G1/S transition, but conclude that, on average, phosphatase identity has an effect on the timing of CDK substrate phosphorylation at the G2/M transition. We have now stated this more clearly in the text (line 276-280).

Can the authors provide information on timing of transition from G2/M after *cdc2* inhibition?

Inhibiting Cdc2 activity partly with 1 μ M 1-NmPP1 leads to an arrest in G2. Washing out 1-NmPP1 releases cells into mitosis, with peak CDK substrate phosphorylation happening 8-12 min after release. The timecourse data in Figure 4 show the second mitosis after the arrest in G2, which happens around 100 min after release from the G2 arrest. We are sorry for the confusion that these data might have caused. We realised that our y-axis labels in Figure 4a got deleted when converting the Figure to a PDF. We have now added a label to indicate the timings of different cell cycle stages under Figure 4a. We have furthermore extended the figure legend and added a schematic of the experimental setup to Supplementary Fig. 10a.

3) Same results section 1: It is unclear to us how these degron cells were generated. Can the authors provide a better description in the materials and methods? provide a reference to the provided table with all strains for clarity.

We have extended our methods section to describe how the degron strains were made and referenced the Supplementary Table 4 including all strains (line 468-470). In brief: a TIR1 base strain was used (Watson *et. al.*, 2021), and we tagged the phosphatases of interest with a C-terminal sAID tag. For phosphoproteomic experiments, these strains were crossed with a *cdc2-as* strain.

4) The schematic in fig1a should be corrected to reflect the actual experimental flow, where *cdc2-as* inhibition takes place first, and then degradation of the phosphatase is achieved, followed by release (this is how we understand it from the extended data 1b panel and as this is more complete, we assume this is the correct version).

We have now amended the figure and stated in the figure legend of Figure 1a that first cells were treated with 1 μ M 1-NmPP1 to induce a G2 arrest, 2.5 hours later auxin was added to degrade the phosphatases of interest. 10 mins after auxin-addition cells were washed into 1-NmPP1-free media (containing auxin) and at peak mitosis CDK activity was fully inhibited by the addition of 10 μ M 1-NmPP1.

5) Line 133: 'countering the notion that phosphatases are unspecific'. This claim lies long behind us and should be rephrased.

This has been rephrased (line 146-148).

6) Section starting at line 202: Could you perform this analysis for CDK-substrates alone as well?

We thank the reviewer for this suggestion and have now also performed this analysis for each phosphatase just for their respective CDK substrates. As expected, CDKs were among the top 10 enriched kinases for all CDK-dependent phosphatase substrates. These data have now been added to Supplementary Fig. 9a.

7) Line 220: instead of 'data show', this should be 'data further confirm', since this is not new. We have changed the wording accordingly (line 251).

8) Section starting at line 222: there is a lot of switching between 'median phosphorylation', 'net phosphorylation' and 'relative phosphorylation'. Are they pointing to the same? If yes, choose one. In any case, they should be defined better, for example, how is relative phosphorylation, used in graphs, calculated?

We have made text changes to clarify the differences between these terms. In brief, we use 'net phosphorylation' to refer to the overall amount of phosphorylated substrate, determined by the catalytic activities of both the kinase and the opposing phosphatase (line 63-65). We use the term "relative phosphorylation" to specifically refer to our phosphoproteomic data, which corresponds to the phosphorylation level normalised to its respective t_0 . In the case of the timecourse phosphoproteomic data, 'relative phosphorylation' refers to phosphorylation level, relative to phosphorylation at peak mitosis. These descriptions have now been added to the methods section (line 733-735) and at the first point of use in the legend of Figure 1d and 4a.

9) Line 232: why data cropped at 40 min? Would be informative to see the entire profile.

The data was cropped at 40 min to focus on the first cell cycle after the release from the G2-block at timepoint 0. The full uncropped data, also showing the first mitosis, has now been added to Supplementary Fig. 10b. It should be noted that the first mitosis happens with high levels of CyclinB (Cdc13) that accumulates during the G2 arrest and is therefore less physiologically informative compared to the second mitosis after the arrest.

10) Line 239: we understand the authors use AUC as a proxy for timing of onset, but this should

be made more clear in the text and not just in the figure legend. Also, can the distribution of relative phosphorylation be shown instead of a single line?

We have now changed Figure 4a to show median and IQR. We also show the distribution of phosphorylation timings in the plot of all individual AUC values in Figure 4b. We have stated in the text that we use the AUC (integral of phosphorylation level between 50-100 min) as a proxy for the timing of CDK substrate site phosphorylation (line 268-271), which is also explained with a schematic in Supplementary Figure 10c.

11) Line 244: S-phase substrates. Where is this data shown? This could be effects of cdc2-as inhibition on G1/S transition.

We have now added a list of S-phase substrate sites, which are targeted by different phosphatases to Supplementary Table 1.

12) Line 268: Please swap legend order of CDC14 and B55 for consistency

We thank the reviewer for noticing this and have swapped the legend order of Supplementary Figure 10e accordingly.

13) Line 271: Please revisit this claim. B55 substrates are more affected in nucleus and at SPB as compared to cytoplasm, as is evident from extended data fig 6b.

The data in Extended Data Figure 6 (now Suppl. Fig. 9) show that in the SPB, cytoplasm and nucleus PP2A-B55 substrates are net phosphorylated on average later than CDC14 substrates. Given the reviewer's concern, we deleted the claim in line 271: "This reinforces the notion that the differences in phosphorylation timing of CDK substrates categorised by phosphatases or subcellular localisation act independently of one another".

14) Also, please describe abbreviations (e.g. SPB) – would be good if authors could double check this throughout text in general.

All abbreviations are now checked and defined at the first point of use.

15) Section starting line 279: we appreciate this section contributes to the relevance of cut3 as a novel b55 substrate. An important shortcoming here is that for b55 the auxin inducible system was used, and for the other phosphatases stable deletion lines were used. Can the authors elaborate on the effect on timing of fraction of cells in peak N/C cut3-mcherry as seen for the other phosphatases.

We used a B55-degron strain in this experiment, since the B55-deletion strain has substantial morphological and growth defects. We have now also analysed SynCut3-mCherry phosphorylation in a B55-delete background and added it to Supplementary Figure 11 b-d, line 327-328. This showed that both degradation and deletion of PP2A-B55 advanced phosphorylation of Cut3-T19, with the effect being slightly more pronounced in the deletion strain. This is likely because deleting the phosphatase abolishes PP2A-B55 activity completely, while degrading B55 may still leave some residual activity. Thus, if at all, the effect seen in the absence of B55 in the degron strain might be a slight underestimation of the effect.

We quantified the time between the initial increase in N/C ratio >1.5 and the peak N/C ratio for Syncut3-mC in the presence and absence of different phosphatases (Figure 5e, Supplementary Fig. 11g). In the absence of B55 or Ppa2, Syncut3-mCherry becomes phosphorylated earlier, and thus the time between initial increase (N/C ratio > 1.5) and peak N/C ratio takes longer. This is not the case in the absence of the other tested phosphatases (line 325-331). In the absence of the Cdc14 and Sds21, the time from initial increase to peak N/C ratio is marginally shorter compared to the control. This may be due to a negative effect of these phosphatases on CDK activity (line 1087-1091).

16) Can authors identify a direct interaction between B55 and cut3?

We conducted a new co-immunoprecipitation experiment, in which we pulled down GFP-tagged Cut3. Since phosphatase-substrate interactions are very fleeting, it is challenging to identify phosphatase substrates by co-immunoprecipitation. Thus, we used very mild conditions for the wash (see Methods). Our new data show that the PP2A catalytic subunits (Ppa2, Ppa1) are significantly enriched in the Cut3 fraction. These new data have now been added to Supplementary Fig. 11a and are mentioned in the text in lines 318-320.

17) Section starting at line 336: Please rephrase this section to be clearer.

As it becomes clear from this data, b55 degradation has a stronger effect than cdc14, and therefore accelerates entry into mitosis faster than B55. If B55 acts later than cdc14, as the authors state, but the effect on B55 is larger,

The size of the effect (extent of length reduction at septation) and timing of the effect are not correlated. This experiment is done in an asynchronous culture. In *S. pombe* the majority of cells are in G2 and thus the later after the degradation of the phosphatase the reduction in length starts, the earlier this effect occurs during the cell cycle. We have now deleted this section for clarity.

18) Line 366: Secondary effects on phosphorylation should be considered.

We have now amended the text to highlight this possibility in the discussion (line 392-396).

19) Line 430: this should be rephrased. Net phosphorylation is affected, not just phosphorylation by cdk.

We thank the reviewer for highlighting this, and have now amended the text accordingly (line 454).

Reviewer #2 (Remarks to the Author)

The cell cycle is tightly regulated by the coordinated actions of cyclin-dependent kinases and their counteracting phosphatases. While CDK-cyclin complexes phosphorylate numerous substrates in a cell cycle-dependent manner, phosphatases play a crucial role in reversing these modifications to ensure proper cell cycle progression. Despite extensive research on CDK-cyclin regulation, the mechanisms by which phosphatases fine-tune the timing and specificity of CDK-mediated phosphorylation remain poorly understood.

To address this critical gap in knowledge, the authors employed *Schizosaccharomyces pombe* as a model system, using phospho-proteomics to systematically analyze phosphorylation dynamics following perturbation of the kinase Cdc2 and four phosphatases: PP2A-B55, PP2A-B56, PP1, and CDC14. Their study successfully identified a broad spectrum of phosphatase-specific substrates, demonstrating that each phosphatase targets distinct pools of Cdc2-phosphorylated proteins. Furthermore, the authors characterized the sequence features of phosphatase-targeted phosphosites, providing mechanistic insights into substrate recognition. Notably, their findings suggest that phosphatases serve as an additional regulatory layer, modulating CDK phosphorylation thresholds required for mitotic entry.

The study is technically rigorous, with well-executed experiments and appropriate data interpretation. However, the following revisions could further strengthen the manuscript.

1) To confirm the robustness of the phospho-proteomic data, Western blot analysis of known Cdc2 and phosphatase substrates should be included. This would provide validation of phosphorylation changes observed in the mass spectrometry experiments.

We thank the reviewer for their comments. The robustness of our phosphoproteomic data analysis to identify Cdc2/CDK substrates has been validated by a previous study from our lab using Western blots against Dis2-T316 (phosphospecific antibody) and by gel-shift for phosphorylations on Orc2 and Sld3 (Swaffer *et. al.*, 2016, Fig. S1h). We identify over 90% of CDK substrate sites previously reported by Swaffer *et. al.* in our dataset. We also show by western blotting against a phospho-TPxK antibody that CDK substrate phosphorylation decreases upon CDK inhibition (Supplementary Figure 1c)

We are confident that our experimental approach identified real phosphatase substrates, given that known phosphatase substrates were identified in our assay (listed in Supplementary Table 1). In addition, the phosphatase substrate motifs identified in our study (Fig. 3e-h) are similar to motifs identified in other model organisms or *in vitro* for these phosphatases, further strengthening the conclusion that we are identifying bona fide phosphatase substrates.

2) The study identifies numerous new phosphatase substrates. However, further biochemical validation (e.g., *in vitro* dephosphorylation assays or mutagenesis of phosphosites) would strengthen the biological relevance of these findings. At least one representative new substrate for each phosphatase should be experimentally verified.

Our phospho-proteomic dataset identified Cut3-T19 as a novel PP2A-B55 substrate. The SynCut3-mCherry fluorescent data in Figure 5 and Supplementary Fig. 11 show that this site is phosphorylated earlier in the absence of PP2A-B55 but not in the absence of other phosphatases, validating Cut3-T19 as a novel PP2A-B55 substrate. We had an unpublished project in the lab developing a phospho-specific antibody against Spo15-S121, which we identified as a novel PP2A-B55 substrate in our phosphoproteomic dataset. Western blotting analysis showed that, as expected, Spo15-S121 is dephosphorylated more slowly in the absence of PP2A-B55, but not PP1. We have now added these data to Supplementary Fig. 4e-h, line 133-135.

We further identified Cut12-T75 as a PP1 substrate with our phosphoproteomic dataset. Western blot analysis using a phospho-specific Cut12-T75 antibody (Grallert *et al.*, 2013) confirmed slower dephosphorylation in the absence of PP1 (Supplementary Fig. 4c-d, line 133-135). Together, these Western blots highlight that we can be confident in our phosphoproteomic data.

Phosphospecific antibodies, against fission yeast proteins, are generally not commercially available. Thus, we could not source a phosphospecific antibody against a CDC14 or PP2A-B56 substrate.

The reviewer also suggested mutagenesis of the phosphosite of a phosphatase substrate. PP2A-B55 preferentially dephosphorylates pT over pS residues. Mutating the Cut3-T19 site to an S should therefore make it a worse PP2A-B55 substrate. Consistent with this, a recent paper from our lab shows that synCut3-T19S is phosphorylated earlier and less switch-like during the cell cycle (Kapadia & Nurse, 2025, Figure 1c&f shown below). Left: green line CytCDK (T19) vs right: green line CytCDKV2 (T19S). We have now added this to the text (line 333-340).

3) To reinforce the claim of phosphatase-substrate specificity, biochemical assays (e.g., phosphatase inhibition or substrate competition experiments) should be performed on selected substrates.

Purifying the investigated phosphatases and example substrates and setting up *in vitro* experiments for them would be a major undertaking and beyond the scope of this paper. We are reassured by our data showing that the amino acid preferences of the investigated phosphatases are very similar to those identified *in vitro* and in other organisms. Phosphatase substrate specificity has been well established *in vitro*, yet whether the same holds true *in vivo* is still a contentious question. The strength and novelty of our paper lie in the direct comparison of multiple phosphatases *in vivo*, as well as highlighting the regulatory consequences of this for the cell cycle. We have highlighted this with text changes in the introduction and throughout the manuscript.

Reviewer #3 (Remarks to the Author):

Zeisner et al engineer *S. pombe* lines with ATP-analog alleles of Cdc2 and AID-tagged alleles of either B55, B56, PP1, and Cdc14. They perform time course phosphoproteomics analysis upon inhibition of Cdc2/CDK and fit their data to an exponential decay curve to identify Cdc2/CDK substrate phosphorylation sites and their opposing phosphatase. They identify kinase-phosphatase interactions and find little site overlap due to differences in phosphatase amino acid preferences for dephosphorylation sites and SLiM recruitment. Furthermore, they temporally order Cdc2/CDK opposing phosphatase activities and determine that each sets a distinct phosphorylation threshold coordinating specific Cdc2/CDK-driven events during mitotic entry. This is an important addition to the regulatory model of mitotic entry, as it more explicitly defines the contribution of ordered phosphatase inactivation to the increase in phosphorylation at the G2/M transition.

Overall, this is a comprehensive and well-carried-out analysis of the Cdc2/CDK opposing phosphatases in *S. pombe* that greatly adds to our understanding of these phosphatases and their contribution to cell cycle regulation. The authors investigate B55, B56, PP1, and Cdc14 under the same experimental conditions, allowing for a robust comparison of datasets to determine phosphatase specificity based on the dephosphorylation of CDK-dependent and -independent sites. Results obtained from these studies reproduce observations made in human cells, demonstrating the conservation of the mechanisms underlying phosphatase specificity and substrate recognition. The time course analysis of dephosphorylation increases the power of the analysis, allowing them to measure the impact on the timing of Cdc2/CDK phosphorylation.

Extended data 1. D/E: Why did the authors observe such highly variable numbers of phosphorylation sites in each of the four phosphoproteomic experiments?

The four phosphoproteomic datasets were processed at different times, which could have contributed to the variability in the overall number of detected sites. However, each of these sets contained the same control samples, which showed that the dephosphorylation rates of CDK substrates in the control condition are highly correlated between experiments (Supplementary Fig. 1l), making us confident that all four phosphoproteomic experiments produced high-quality reproducible data.

1F: How many sites did the authors detect in the proline-directed versus non-proline-directed category?

Across all four phosphoproteomic-experiments we identified 1707 CDK substrates sites. Of these sites 705 were non-proline sites (S/T nonP, Supplementary Fig. 1h). These n-numbers have been added to the figure.

Extended data 2B. The differences in pTPxK levels are not noticeable. Quantification and normalization to ponceau staining would help to visualize these differences.

We used the phospho-TPxK antibody (Supplementary Fig. 1c) as a control to test whether CDK activity had been inhibited in our experiment before proceeding with the phosphoproteomic workflow. As expected, a decrease in band intensity can be seen comparing timepoints 0 and 25 minutes. We have now stated this in the figure legend of Supplementary Fig. 1c and reordered the figures to make the purpose of these Western blots clearer. We thank the reviewer for highlighting this.

As suggested, we have now also quantified these Western blots, which indicated that the 120 kDa phosphopeptide is likely dephosphorylated by PP1, and the 70 kDa phosphopeptide may

be dephosphorylated by PP2A-B55. We also did an additional Western blot in the absence of PP2A-B55, to confirm this (shown below). It should be noted that it is not known which peptide sequences correspond to these bands.

The authors state that: “they classified CDK substrate sites as phosphatase targets if their dephosphorylation was substantially slower in the absence of one of the phosphatases”. Here, clearly stated cutoffs should be used. Was any statistical test performed?

We described the phosphatase substrate identification in detail in the methods section and Supplementary Fig. 1d-e. We have now extended the methods sections to include all information in one place, including information on the statistical tests used (line 584-598).

In addition to their scheme, the authors should include examples of known B56 or Cdc14 CDK-independent substrates in Figure 3a.

We agree with the reviewer that examples of phosphatase substrates would work well in Figure 3 and have now added example substrate sites of PP2A-B56 (Rgf2-S746) and CDC14 (Isd90-S99) to Main Figure 3b.

A bar graph directly comparing the number of CDK-dependent and -independent substrates identified across all experiments for phosphatase should be included in the main text Figure 3. We have now added a bar graph comparing the number of CDK-dependent and -independent substrates identified across all experiments in Figure 3c.

The data shown in extended data Figure 5c for Cdc2/CDK-independent substrates seems high for PP2A-B55 and PP1. Do the authors have an explanation for this?

Overall, PP2A-B56 and PP1 have the largest number of identified CDK-independent phosphatase substrates, with 1280 and 1121 substrate sites, respectively (Supplementary Fig. 8b). This is partly due to differences in the overall number of identified phosphatase substrates

in the four experiments. We have now added a figure to depict the % of CDK-independent substrates as a percentage of all sites identified in the respective experiment (Supplementary Fig. 8d). This shows that PP1 and PP2A-B55 target more CDK-independent phosphatase substrates than PP2A-B56 and CDC14. PP1 is a holoenzyme, so by degrading its catalytic subunits (encoded by *dis2* and *sds21* in fission yeast), we are inhibiting all PP1 enzymes, which together are expected to have more substrates than a single phosphatase such as CDC14. PP2A-B55 is one of the main cellular phosphatases, and thus, we would expect that it targets more sites than CDC14.

Representing these sites as weblogs, in addition to the ice logo, and performing a motif enrichment analysis would provide additional information on the type of sites dephosphorylated by each of the phosphatases.

We thank the reviewer for this suggestion; we have now added Weblogs and frequency logos of CDK-dependent (Supplementary Fig. 6a,b) and CDK-independent phosphatase substrates to the manuscript (Supplementary Fig. 8f,g). We furthermore conducted a motif enrichment analysis using the MEME tool suite SEA analysis (Bailey & Grant, 2021; Bailey *et al.*, 2015). This analysis yielded between 7 and 23 motifs (5-15 amino acids long) enriched in each of the phosphatase substrate sites. Interestingly, CDC14 substrate sites were enriched for SILK motifs, a known PP1 SLiM, suggesting that PP2A-B56 substrate sites may be in close proximity to PP1 binding sites (line 240-246). All identified motifs have been added to a new Supplementary table 3.

The authors' analysis would be more impactful if they could perform an additional phosphoproteomics time course analysis for a different kinase for which they predict a specific phosphatase opposition course analysis.

We re-analysed the phosphoproteomic timecourse data, specifically looking at Polo and Aurora sites, to test whether a) these kinase sites are also opposed by multiple phosphatases and b) whether there are differences in phosphorylation timing according to which phosphatase targets the kinase site. We identified between 39 Aurora and 50 Polo substrate sites in the timecourse data (see figure below). Of these sites, we identified an opposing phosphatase for 5 and 9 sites. Due to the low n-numbers, we could not make conclusions about whether Aurora or Polo kinase sites opposed by different phosphatases are net phosphorylated at different points during the cell cycle.

The timing analysis is very interesting. The authors infer phosphatase activity based on when their substrates are phosphorylated by Cdc2/CDK in G2/M. However, the inferred effect on phosphatase activity is never stated. It would help the reader if the authors would do so and, in addition to their diagram of Cdc2/CDK activity thresholds, provide a model for phosphatase activity levels in G2/M.

We agree and highlight in our discussion that the differences in timing between CDK substrates opposed by different phosphatases may be due to a sequential inactivation of these phosphatases at the G2/M transition (line 423-429). Alternatively, different phosphatases may oppose CDK substrates with different strengths (line 418-421). We have now made text changes and added these possible mechanisms as schematics to the model in Figure 7b.